# Assessing internal changes in the future structure of Dry-Hot compound events. The case of the Pyrenees

Marc Lemus-Canovas[1], Joan Albert Lopez-Bustins[1]

[1]Climatology Group, Department of Geography, University of Barcelona, Barcelona, 08001, Spain

*Correspondence to*: Marc Lemus-Canovas (mlemus@ub.edu)

**Abstract.** Impacts upon vulnerable areas such as mountain ranges may become greater under a future scenario of adverse climatic conditions. In this sense, the concurrence of long dry spells and extremely hot temperatures can induce environmental risks such as wildfires, crop yield losses or other problems, the consequences of which could be much more serious than if these events were to occur separately in time (e.g. only long dry spells). The present study attempts to address recent and future

changes in the following dimensions: duration (D), magnitude (M) and extreme magnitude (EM) of compound Dry-Hot events in the Pyrenees. The analysis focuses upon changes in the extremely long dry spells and extremely high temperatures that occur within these dry periods, in order to estimate whether the internal structure of the compound event underwent a change in the observed period (1981-2015) and whether it will change in the future (2006-2100) under intermediate (RCP4.5) and high (RCP8.5) emission scenarios. To this end, we quantified the changes in the temporal trends of such events, as well as

changes in the bivariate probability density functions for the main Pyrenean regions. The results showed that to date the risk of the compound event has increased by only one dimension –magnitude (including extreme magnitude) – during the last few decades. In relation to the future, increased in risk was found to be associated with an increase both in the magnitude and the duration (extremely long dry spells) of the compound event throughout the Pyrenees during the spring under RCP8.5 and in the northernmost part of this mountain range during summer under this same scenario.

## 1.  Introduction

Research on dry spells or droughts, as well as extreme heat events (i.e. heat waves) is habitually based upon an individual focus, and the compound nature of such events is often neglected. In this sense, in the case of spells (whether dry or wet), several studies have examined the duration thereof and have quantified the trends of such events in different regions of the

world (Martin-Vide and Gomez, 1999; Zolina et al., 2013; Singh et al., 2014); however, the thermal component of these episodes has not been addressed therein. Similarly, we found different studies on temperature extremes that did not evaluate the effect of duration of such extremes (Diffenbaugh and Ashfaq 2010; Fonseca et al., 2016; Salameh et al., 2019). In general terms, the indices proposed by the Expert Team on Climate Change Detection and Indices (ETCCDI) do not involve analyzing events in a compound manner, a shortcoming that can result in underestimation of risk (Zscheischler et al., 2018).

As mentioned above, compound analysis of events enables us to estimate the real risk induced by the simultaneous occurrence of several climatic variables; this is of particular interest in fragile and vulnerable areas, such as mountain ranges, in a context of anthropogenic climate change. In this sense, the area of the Pyrenees (Fig. 1), a transboundary area between three countries (Andorra, France, and Spain), possesses a great wealth of natural resources and a high level of biodiversity. However, some studies have already addressed the initial impacts of the warming observed in this region, particularly in relation to the decline

of mountain forests (Camarero 2017). In addition, more frequent dry periods and droughts have also led to the defoliation of silver fir (Abies alba) forests in this region (Gazol et al., 2020). There is therefore an urgent need for a compound analysis of extreme dry spells and extreme warm temperature events, i.e. the combination of duration (D) and magnitude (M), as conceptualized in Manning et al., (2019), in order to ascertain whether these compound events will be more widespread in the future and whether they pose other risks such as wildfires or crop yield losses.

As for the compound analysis of Dry-Hot events, previous studies have highlighted an increase in the frequency and spatial scope of such events in recent years in several areas such as the US (Mazdiyasni and AghaKouchak 2015), India (Sharma and Mujumdar 2017) or China (Wu et al., 2019), although in Europe the magnitude (temperature) of these events was revealed to have greater weight than their duration (dry spells) as indicated by Manning et al., (2019). Several recently published studies focus mainly on the changes observed in these compound events (Wu et al., 2019; Hao et al., 2019; Manning et al., 2019).

However, fewer analyses have employed future projections to assess the risk posed by the occurrence of compound events (Zscheischler and Seneviratne 2017; Lu et al., 2018; Wu et al., 2020). Zscheischler and Seneviratne (2017) used Copula's method to evaluate changes in the probability of future Dry-Hot compound events; they employed the Coupled Model Intercomparison Project (CMIP5) simulations to show an increase in the probability of these events in most regions of the world. Previous research has therefore highlighted a general increase in this kind of events, but they have neglected to

separately address the importance of the variables contributing to such compound events. Thus, the present study will attempt to account for the influence of the variables giving rise to Dry-Hot compound events. Herein we analyze the observed and projected changes in Dry-Hot compound events, understanding these as the combination of extremely long dry periods and extremely high temperatures within these periods.

Bias correction techniques are used to correct the data simulated by global and regional climate models (GCM and RCM) by

means of observed data. Indeed, these techniques are most commonly employed to correct for only one variable at a time (Teutschbein and Seibert, 2012; Rajczak et al., 2016). Nonetheless, a univariate correction can moderately affect the mutual structural dependence of different variables (Wilcke et al., 2013), e.g. temperature vs. precipitation, although recent studies have shown that univariate bias correction methods can be sufficiently robust for certain specific regional impact studies (Casanueva et al., 2018). However, in the treatment of compound events, the use of multivariate bias correction methods can

provide an added value (François et al., 2020) by optimally estimating multivariate dependence.

The main objectives of our paper are: 1) to characterise the duration (D), magnitude (M) and extreme magnitude (EM) of events; 2) to estimate the observed regional trends of the variables at play in the compound event; 3) to project the future trends

of such compound events, under different Representative Concentration Pathways (RCPs), in order to determine future changes in the weights of each variable involved in the compound event. As an intermediate and essential step between tasks 2 and 3, we will apply and evaluate a bias correction in relation to the historical simulations.

Section 2 describes the data collection method used in our study, including the observed and simulated data, the methodology employed to obtain the regionalized series of the Pyrenees, the criteria used to define each event, and the bias correction method and the assessment thereof. In section 3, we present the exploratory analysis of the variables constituting the compound event, as well as the observed trends in the main regions of the study area defined in section 2. In section 4, we perform an exhaustive evaluation of the effects of the bias correction methods applied to the series simulated by the climate models. Subsequently, in section 5, we analyse the results of the projection of the variables described in section 2 with the use of different RCPs, in order to estimate changes in the internal structure of the Hot-Dry compound event. Finally, section 6 presents the discussion of the results, and section 7 some brief conclusions.

## 2. Data and methods

### 2.1 Observed and projected data

The accumulated daily precipitation and maximum daily temperature of the spring and summer seasons (MAM and JJA) were extracted from the database of the CLIMPY project (Characterization of the evolution of climate and provision of information for adaptation in the Pyrenees); this is a transboundary initiative whose objective is to perform a detailed analysis of recent trends in temperature, precipitation and snow cover in the Pyrenees, as well as the future projection thereof (Cuadrat et al., 2020). These two variables were provided in a 1x1 km high-resolution grid, for the 1981-2015 period, and fed by 1,343 weather stations located in Andorra, France and Spain; the grid was built following the quality control, reconstruction and gridding processes according to Serrano-Notivoli et al., (2017 and 2019). We focused on spring and summer, as spring can constitute the precursor of summer wildfires (Turco et al., 2017), and is a season prone to crop yield losses (Zscheischler et al., 2017), etc. We also centred our attention on summer, as this is the hottest and driest time of year in the area and is the most critical period for the occurrence of the above mentioned environmental risks.

We used the 850 hPa temperature (T850) and the 500 hPa daily geopotential height (Z500) from ERA-Interim (Dee et al., 2011) at a spatial resolution of 0.75° to synoptically characterize these compound Dry-Hot events. For the climate simulation projections, six climate models were obtained from different Regional Climate Models (RCMs), which were nested in different General Circulation Models (GCMs) and computed over Europe (Table 1), within the framework of the Coordinated Regional Climate Downscaling Experiment (EURO-CORDEX) (Jacob et al., 2014). These gridded projections cover all of Europe with a spatial resolution of 0.11° in latitude and longitude (~12 km) for the 1981-2005 (historical experiment) and 2006-2100 (RCPs scenarios) periods. We selected the climate models that provided sufficient data for such a study, and which had been used in previous research (Jacob et al., 2014). Additionally, the RCPs used were the 4.5 –stabilization without overshoot pathway to 4.5 W/m2 (~650 ppm $CO_2$) stabilization after 2100 (Wise et al., 2009)– and the 8.5 –rising radiative forcing pathway leading

to 8.5 W/m2 (~1370 ppm $CO_2$) by 2100 (Riahi et al., 2007)–. Herein we did not employ the gridded data but rather the cell closest to the centroid of each region (Fig. 1), in order to avoid inflation issues and misrepresentation of subgrid variability when bias correction methods (Maraun et al., 2013, 2017) were used.

**Table 1. EURO-CORDEX climatic models used and their characteristics. Source: Copernicus Climate Change Service (https://cds.climate.copernicus.eu/cdsapp#!/home, last accessed 30 April 2021).**

| Model | Institute | GCM | RCM |
|---|---|---|---|
| 1 | CNRM | CNRM-CM5 | ALADIN-63 |
| 2 | CNRM | CNRM-CM5 | RACMO22E |
| 3 | DMI | NCC-NorESM1-M | DMI-HIRHAM5 |
| 4 | KNMI | EC-EARTH | RACMO22E |
| 5 | SMHI | IPSL-CM5A-MR | RCA4 |
| 6 | SMHI | MPI-ESM-LR | RCA4 |

**2.2 Regionalization**

The Pyrenees constitute a mountainous system presenting high climatic variability, which can be summarized quite easily in
order to explain the major part of the compound behaviour of Dry-Hot events. In this sense, the authors consider that to divide Pyrenees in many different regions is of no particular interest for the present analysis, because situations of long dry spells and extremely hot temperatures, for instance, display a practically identical synoptic behaviour pattern throughout the region. For example, a subtropical ridge produces a dry environment and above-average temperatures throughout southern Europe (Sousa et al., 2018), and hence in the Pyrenees (Lemus-Canovas et al., 2019a). This does not occur when spatial patterns of
precipitation are investigated, because spatial variability is much greater. Interestingly, with northern advection in this area, precipitation can be abundant on the Atlantic and northern slopes, but scarce or non-existent on the southern slopes (Lemus-Canovas et al., 2018). This variability therefore differs depending on the variables analyzed.

Although the synoptic driver of these dry-hot compound situations is broadly the same for the whole Pyrenees region, several geographical factors such as altitude, latitude or distance from the sea endow these events with different degrees of intensity
in different areas of the Pyrenees. This subtropical ridge will have a greater impact in the southern area of the Pyrenees than

in the northern sector, simply because of its proximity to the origin of the subtropical air mass. It is therefore of great interest to divide the Pyrenees into a series of basic regions exhibiting relatively different behaviour patterns.

The use of clustering techniques is very common in the creation of regions of climate variables. For example, Carvalho et al., (2016) regionalized temperature and precipitation in Europe; Carro-Calvo et al., (2017) performed similar tasks for tropospheric ozone; and more recently, Lemus-Canovas et al., (2019b) employed these techniques by combining precipitation with circulation types to establish rainfall regions in the Alps. In the present paper we conducted a combined regionalization of temperature and precipitation, (as both variables constitute the basis of Dry-Hot events) by applying the k-means algorithm to the daily series of temperature and rainfall (normalized) of spring and summer. In order to obtain a robust regionalization, we established a maximum of 100 iterations and 300 repetitions. As the algorithm converges, the spatiotemporal patterns of temperature and precipitation for all cells are observed to be consistent within a given region (i.e. close to the centroid), and each grid cell is seen to be better represented by its cluster centroid than by any other centre of cluster. To decide the optimal number of clusters, we performed an iteration from k = 2 to k=15, obtaining 14 different classifications (see Fig. S1 in the Supplement). In order to establish the explained variance (EV) for each new created region (Eq. 1), we computed the mean of squared distances between cluster centres (*betweenss*) and the total sum of squares (*totss*). The highest possible values are expected, since the aim involves achieving a clear separation between clusters. Totss is the sum of *betweenss* and total *withinss*. *Withinss* is the within cluster sum of squares. It results in a vector with a value for each cluster. The lowest possible values are expected, since homogeneity within the clusters is sought.

$$EV = \frac{betweenss}{totss} \cdot 100 \qquad \text{(Eq. 1)}$$

The percentage of explained variance can be explained by the increase in k clusters, as shown in the Scree test (Cattell, 1966) in Fig. S2. Such a representation shows two points, – k = 5 (40%) and k = 8 (48%) – which could be considered as a "slope change", and therefore possible delimiters of the number of regions. Despite the use of the Scree test, the decision is subjective, and a compromise is therefore needed between the degree of complexity and the descriptive capacity of the regionalization (Carro-Calvo et al., 2017). Consequently, we decided to use 8 clusters, which explain 48% of the variance (Fig. S2).

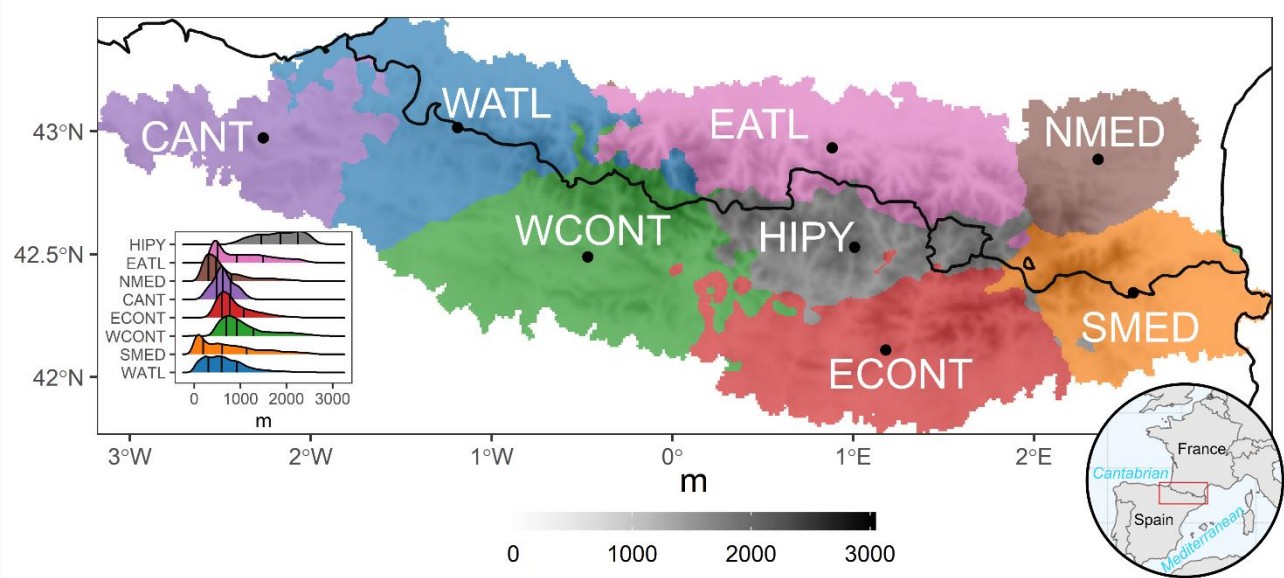


**Figure 1. Regionalization obtained with the k-means method. From left to right: CANT: Cantabrian; WATL: West Atlantic; WCONT: West Continental; EATL: East Atlantic; HIPY: High Pyrenees; ECONT: East Continental; NMED: North Mediterranean; SMED: South Mediterranean. In the bottom left-hand corner, the elevation probability density curves (m) are shown for each region. The vertical lines indicate the 25th 50th and 75th quantiles. The elevation base map was generated using the**
**data provided by the Shuttle Radar Topography Mission (SRTM).**

For the construction of the regionalized series, the daily values of all cells were averaged in order to work with a series that is smoother than if the centroids were used. The main reason for working with averaged regional series was to avoid the downscaling process in the application of the bias correction method. Thus, inflation and modification of the trend represented
by the climate model (Maraun 2013) were avoided, among other undesirable effects. This and other aspects relating to the application of the bias correction are explained in section 2.4.

### 2.3 Event definition

As previously stated, the Dry-Hot events are characterized by means of the following dimensions: duration (D), magnitude (M) and extreme magnitude (EM), corresponding to the spring months: March, April and May (MAM); and to the summer
months: June, July and August (JJA); both seasons are analyzed independently. D is defined as the number of consecutive days on which precipitation is below 1 mm (Fig. 2). This threshold was chosen to be consistent with previous studies (Orlowsky and Seneviratne 2012; Donat et al., 2013; Lehtonen et al., 2014; Manning et al., 2019), as well as to avoid the drizzle effect, which systematically causes climate models to overestimate precipitation (Gutowski et al., 2003).

To ensure that independent and extreme spells are obtained, for each year (spring and summer, separately) we computed the
duration of the 95[th] percentile of dry spells, subsequently selecting the ones displaying a duration longer than this threshold.

We performed a sensitivity test in order to select an appropriate threshold capable of detecting sufficiently long and robust dry spells, especially in wet areas with few dry spells. Additionally, M is the conditional distribution of daily maximum temperatures (Tx) during long dry spells (D) while EM is the conditional distribution of temperature during dry spells that exceed the 95th percentile of daily Tx. Tx above 95th percentile occurring outside of these long dry spells (D) are not

considered. Figure 2 shows the performance of these three variables in a grid cell time-series in the easternmost part of the Pyrenees.

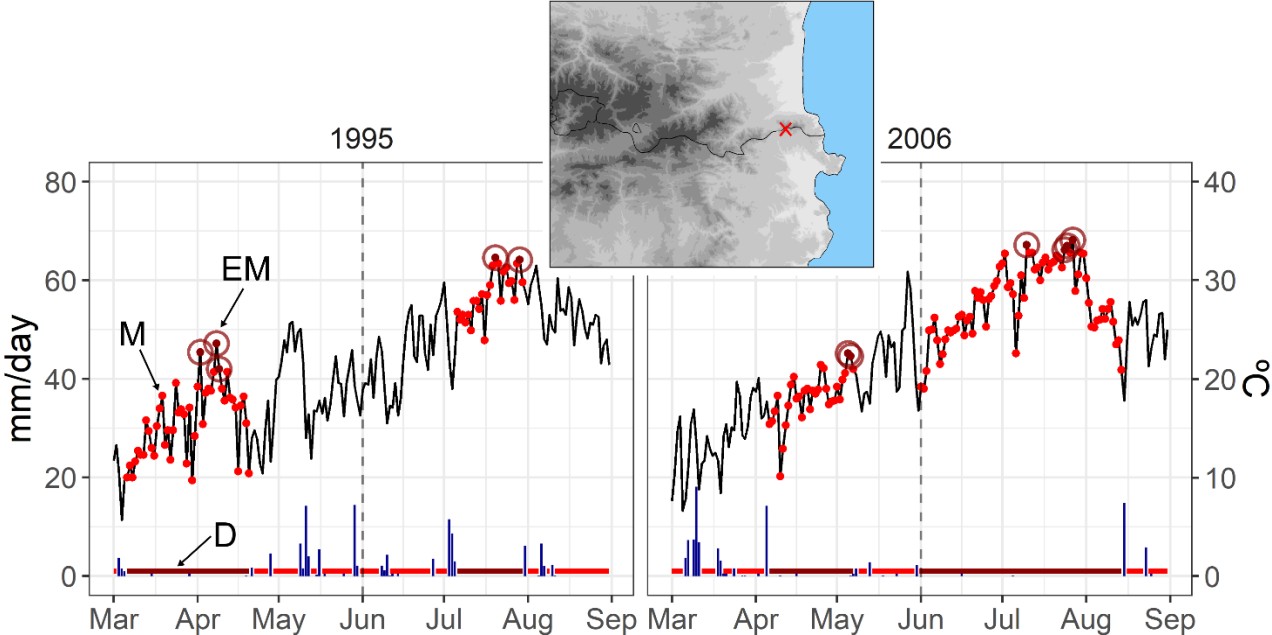

**Figure 2. Time series of daily precipitation (blue vertical bars) and daily maximum temperature (Tx) (black line) at a mountainous location in the easternmost part of the Pyrenees (x = 2.9, y = 42.5). The red horizontal bars at the bottom end of the plot show the**
**length of dry spells, while the dark red bars show the extremely long dry spells (D). The red dots (M) show the Tx values during a D event. The dark red dots with a circle indicate Tx values above the 95th percentile during a D event. The dashed vertical line indicates the separation between the spring and summer seasons. The selection of years highlights a dry spring (1995), and a dry summer (2006) in this area.**

Analyzing the EM subset enables us to characterise the greater risk of the simultaneous occurrence of both variables: D and EM, which in turn may significantly increase the risk of wildfires, for example. To estimate the trend of the events and to assess the statistical significance of these trends we employed Sen's slope (Sen, 1968) and Mann-Kendall's non-parametric test (Mann, 1945). We also computed the annual temperature anomalies of M and EM with respect to the daily Tx mean (spring and summer) over the1981-2015 period, in an attempt to quantify temperature anomaly differences between these two types
of variables.

## 2.4 Bias correction and evaluation

In the present study 1) we calculated the mean of daily Tx and daily precipitation from the observed grid cells belonging to each of the regions of the study area (Fig. 1); 2) we extracted the time series from the cell closest to the centroid of each region for each RCM (Table 1); 3) we applied a univariate and a multivariate bias correction method for correcting daily Tx and daily precipitation from the RCMs; 4) we aggregated to D, M and EM indices the RCM corrected daily Tx and daily precipitation. One of the most popular and widely used techniques for the univariate bias correction is Quantile Mapping (QM). Bias correction by QM is frequently used to downscale simulations at the station level or in high-resolution grid boxes; however, it induces inflation problems in the corrected series (Maraun, 2013) and is unable to generate daily subgrid variability (Maraun et al., 2017). The above-mentioned issues tend to be exacerbated in mountain areas, where many local processes may not be represented following the QM process (Maraun and Widmann, 2018b). As an alternative to univariate bias correction, and in an attempt to correct the inter-variable correlation, different multivariate bias correction methods have been proposed over the last few years (Piani and Haerter, 2012; Vrac and Friederichs, 2015; Cannon 2016, 2018a).

In the present research we first employ a univariate bias correction approach, the empirical quantile mapping (EQM) method, which estimates the values of the empirical cumulative distribution function (CDF) of the observed and modelled time series for each quantile (Panofsky and Brier, 1968; Gudmundsson et al., 2012). Hence, if $X_o$ and $X_m$ are the observed and modeled values, respectively, then:

$$\hat{X}_m = F_o^{-1}\left(F_m\left(X_m\right)\right) \qquad (1)$$

where $F_m$ is the empirical cumulative distribution function of $X_m$ and $F_o^{-1}$ is the inverse empirical distribution function (or quantile function) corresponding to $X_o$. We apply univariate QM with the R package qmap (Gudmundsson, 2014) using 100 quantiles. The drizzle effect was corrected using a wet day threshold of 1 mm/day for the observations (Hay and Clark 2003; Piani et al., 2010)

Second, the multivariate bias correction method used is the one proposed by Cannon (2018): The Multivariate Bias Correction with N-dimensional probability density function transform (MBCn). This method is based on an adaptation of an image processing algorithm used to transfer color information; MBCn enables the statistical characteristics of a reference multivariate distribution to be transferred to the multivariate distribution of climate model variables. The MBCn method can be summarized in four steps with regard to how it corrects climate simulations. In step (a), MBCn uses the quantile-delta mapping method (QDM; Cannon et al., 2015), which preserves absolute or relative changes in quantiles, e.g., for variables such as temperature or ratio variables like precipitation. In step (b), once univariate distributions have been corrected, the dependence structure is adjusted by means of an iterative process *j*. In each step, data are multiplied by random orthogonal rotation matrices to partially decorrelate the climate variables to be corrected.

$$\tilde{X}_{\mathrm{m}}^{[j]} = X_{\mathrm{m}}^{[j]} R^{[j]},$$
$$\tilde{X}_{p}^{[j]} = X_{p}^{[j]} R^{[j]}, \qquad (2)$$
$$\tilde{X}_{o}^{[j]} = X_{o}^{[j]} R^{[j]}$$

In step (c), we apply the absolute change form of QDM to each variable in $\tilde{X}_{\mathrm{m}}^{[j]}$ and $\tilde{X}_{p}^{[j]}$, using the corresponding variable in $\tilde{X}_{o}^{[j]}$ as the target, yielding $\hat{X}_{\mathrm{m}}^{[j]}$ and $\hat{X}_{p}^{[j]}$. In step (d), rotate back:

$$X_{\mathrm{m}}^{[j+1]} = \hat{X}_{\mathrm{m}}^{[j]} R^{[j]^{-1}},$$
$$X_{p}^{[j+1]} = \hat{X}_{p}^{[j]} R^{[j]^{-1}}, \qquad (3)$$
$$X_{o}^{[j+1]} = X_{o}^{[j]}$$

The steps (b)–(d) are repeated until the multivariate distribution of $X_{\mathrm{m}}^{[j+1]}$ matches $X_{o}$. We applied MBCn with the R package MBC (Cannon, 2018b)

Both bias correction methods were evaluated by means of a 7-year 5-fold cross validation of (4 folds for adjustment and 1-fold for validation). Cross-validation should not be applied on validating free-running climate simulations against observed series, as the climate models are temporarily stochastic and could induce serious errors in the assessment of the daily series
(Maraun and Widmann, 2018a). However, herein we work in a seasonal scale -spring and summer- on the variables D, M and EM (see section 2.3), where the RCM is expected to be able to reflect the seasonality component and trend.

We assessed the structural dependence between temperature and precipitation, which was bias-corrected, by means of Pearson's correlation coefficient between the observed series and the historical simulation (uncorrected) series, and between the observed series and the bias corrected historical simulation (corrected). Prior to the correlation, we averaged daily
temperature and daily precipitation to each Julian day for the whole series (1981-2014) in order to avoid noise in the results.

In addition, we tested the differences between the simulated and observed distributions of extreme long dry spells, using the Kolmogrov-Smirnov (KS) test. This test serves to evaluate the weaknesses of bias correction methods in order to accurately estimate the length of dry spells, which were noted in previous research (Rajzak et al., 2016; Maraun et al., 2017). Furthermore, we tested the bias estimation for temperature using two thresholds: the 95[th] percentile of temperature distribution, and the 95[th]
percentile of temperature distribution during the during the 95[th] percentile of extremely long dry spells in order to discuss the performance of each bias correction method in these extreme temperature situations.

## 3. Characterization of the variables underlying the compound event and of the role they play in potential risks

Extremely long dry spells (D) have a main north-south pattern in which the northernmost areas present extreme D values of fewer than 15 days in spring and summer, and the southernmost areas provide values that can exceed 30 days, mainly in

summer (Fig. 3). A second spatial pattern enabled the Atlantic and Mediterranean coastal areas to be differentiated. The former area presented the lowest number of extreme spells throughout the study area in spring and summer. On the other hand, the Mediterranean area showed a very high number of extreme dry spells, especially in summer, when these lasted on average up to 50 days. These spatial patterns showed that, despite the small size of the study area, the D patterns are very diverse.

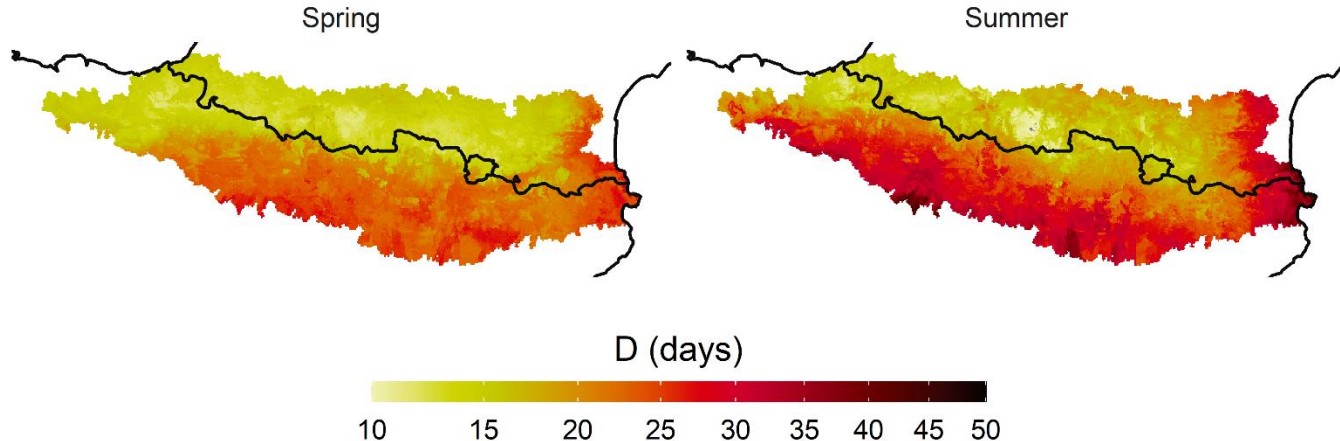

**Figure 3. Annually averaged extremely long dry spells (>95th percentile) for spring and summer during 1981-2015 period.**

However, the present paper did not only focus upon variable D, we also examined the combination of this variable and extremely high near-surface temperature. In this sense, it is important to emphasize the difference between analyzing only the Tx values of the days comprising D, which we called M, and analyzing the Tx values >95th percentile (EM). This difference is illustrated in the Tx anomalies of both periods (Fig. 4).

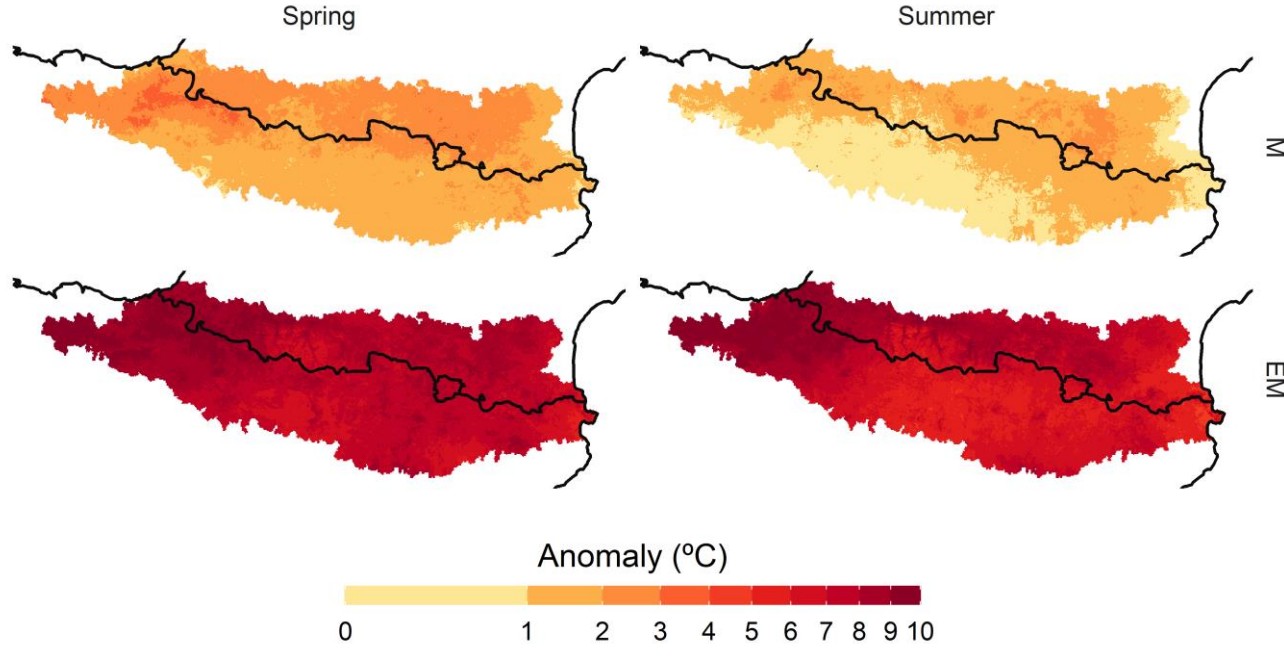

**Figure 4. Annually averaged maximum temperature (Tx) anomaly for M and EM variables and for spring and summer with respect to daily means over the 1981-2015 period.**

Extremely long dry spells (D) are inherently characterised by warmer than normal periods (M). The long dry spells give rise to temperatures between 1 and 4ºC above the mean temperature in spring, and between 0 and 3ºC in summer. Although these anomalous temperatures are not extremely high in the dry periods, especially in summer, if we analyse the thermal extremes (EM) occurring within the D events, in spring and summer the thermal anomaly with respect to the normal values of these two seasons is observed to reach up to 10ºC above the average in the northern half of the study and in the area of Atlantic influence of the study area. In the southernmost region, these anomalies are also accentuated, between 6 and 8ºC above average. The reason why the thermal anomalies are slightly higher in the northern and Atlantic region of the study area is mainly due to the fact that in these areas the number of days with precipitation (and therefore with a moderate Tx) is very high by default (Lemus-Canovas et al., 2019a). Consequently, although the spells are short, they give rise to an extremely positive thermal anomaly, mainly on the hottest days of the spell (See EM for Summer in Fig. 4). In contrast, in the south of the Pyrenees, most days present hardly any precipitation, especially in summer, and dry spells and a positive thermal anomaly are therefore not synonymous (see M for summer in Fig. 4). A similar explanation can be found in the seasonal differences: summer is the dry season in most of the study area, which usually presents high thermal values and no precipitation; consequently, thermal anomalies of M and EM are generally lower than those observed in spring. These surface conditions are also reflected in the upper layers (Fig. 5). Precisely, the mean thermal anomalies at 850 hPa during D events are slightly greater than normal,

between 0 and 2°C above the mean. However, when analysing the set of extreme thermal days (EM) in the D events, the anomalies at 500 hPa are also seen to reach very high values, between 5 and 7°C, just above the study area. It also confirms a greater enhancement of the subtropical ridge in the EM than in the D events.

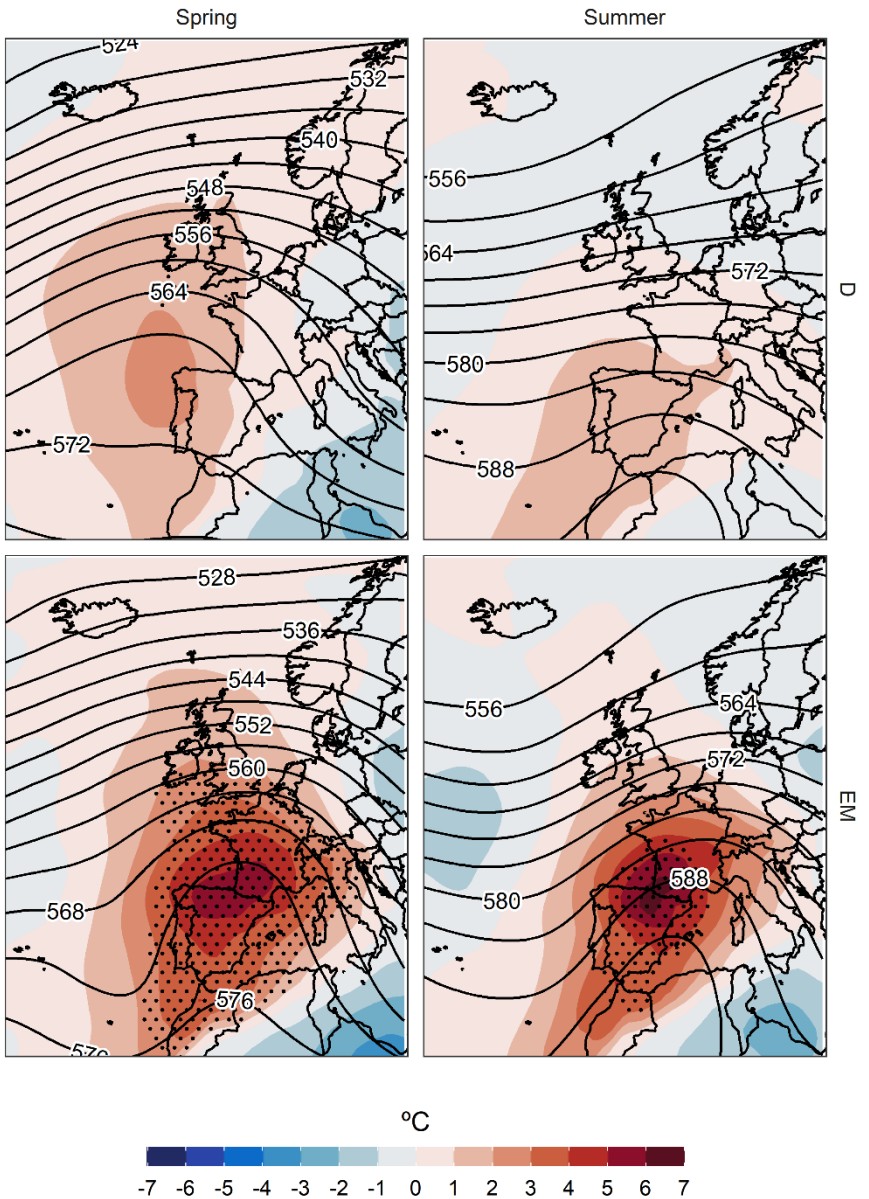

**Figure 5. Daily anomalies of temperature at 850 hPa (°C, shading) and absolute geopotential height at 500 hPa (dm, contours), with contour interval of 4 dm for the days comprising D and EM variables in spring (1322, 104) and summer (1396, 111), respectively, in the HIPY region (most centered region of the Pyrenees). Dots identify regions under flash heating conditions (T850 daily mean above the local daily 99th percentile (with respect to 1981–2015), computed with a 31-day centred window).**

Before analyzing the future projection of the variables D, M and EM, we reviewed the observed trends of such variables for each region and season. In the case of D (Fig. 6a), a non-significant trend was observed (p-value ≤ 0.05) at the 95% confidence level. A high intrannual variability of the duration of extreme dry spells was detected.

This did not occur on assessing the EM and M trends (Fig. 6 c, b, respectively), as both variables displayed a tendency to increase. This trend presented a higher slope in the spring and in the case of the EM. Indeed, the annual values of EM for spring and summer were almost all positive, whilst this was not the case on evaluating only M. At an intra-regional level, the main differences were observed in summer for EM, when the Mediterranean regions NMED and SMED accounted for a higher slope than the other regions. On the contrary, in spring growth was practically the same for all regions for EM and M.

Remarkably, the HIPY region did not show a significant increase in summer for EM and M for a p-value ≤ 0.05.

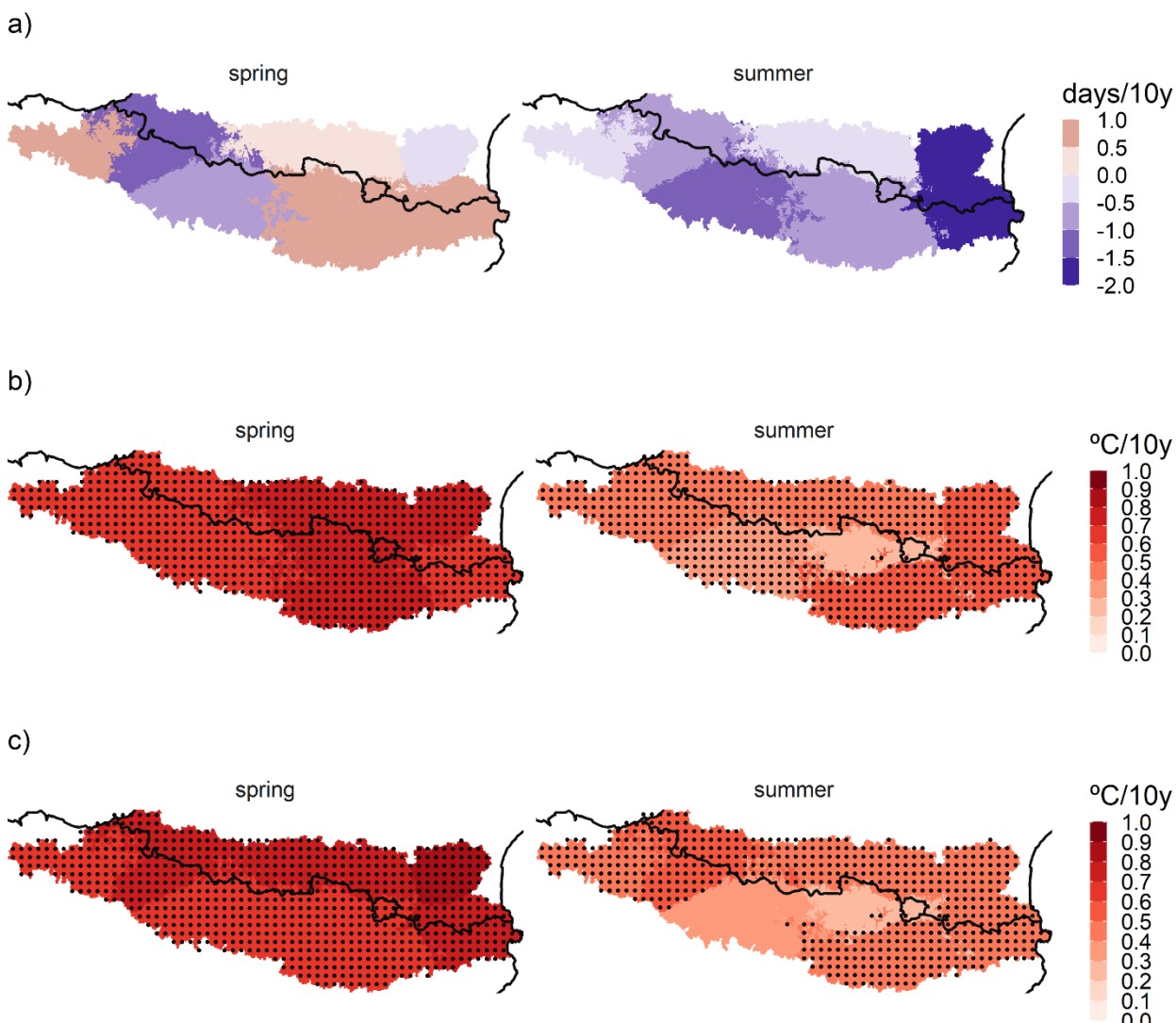

**Figure 6. 10-years trend (Sen's slope) for a) extreme dry spell events per year (D), b) for mean maximum temperature events within the dry event per year (M) and for c) mean extreme maximum temperature events within the dry event per year (EM); for the spring and summer seasons and for all regions of the study area. The stippling shows statistically significant regions at the 95% confidence level.**

## 4. Assessing the reliability of the bias-corrected projections

We evaluated the BC methods in order to estimate 1) how they are able to represent the dependence structure between temperature and precipitation (Fig. 7 & Fig. 8); 2) how well extremely long dry spells are simulated by RCM models, and to

ascertain the contribution of the bias correction methods (Fig. 9); 3) the degree of bias of daily maximum temperatures conditioned to extremely long dry spells (Fig. 10).

Regarding the structural dependence between temperature and precipitation, in the case of the CANT region (Fig. 7) a better correlation was observed in the simulation corrected with the MBCn method (Fig. 7b) in comparison with the UBC method (Fig. 7a). Both methods adjust the bias of the marginal distributions, but the MBCn can reproduce the dependence relationship between precipitation and temperature more closely to the observed values than UBC. A similar situation occurs in the dipole area of the NMED study area (Fig. S3), where MBCn (Fig. S3b) tends to cause an increase in the correlation coefficient between temperature and precipitation, even above the correlation value estimated in the data observed.

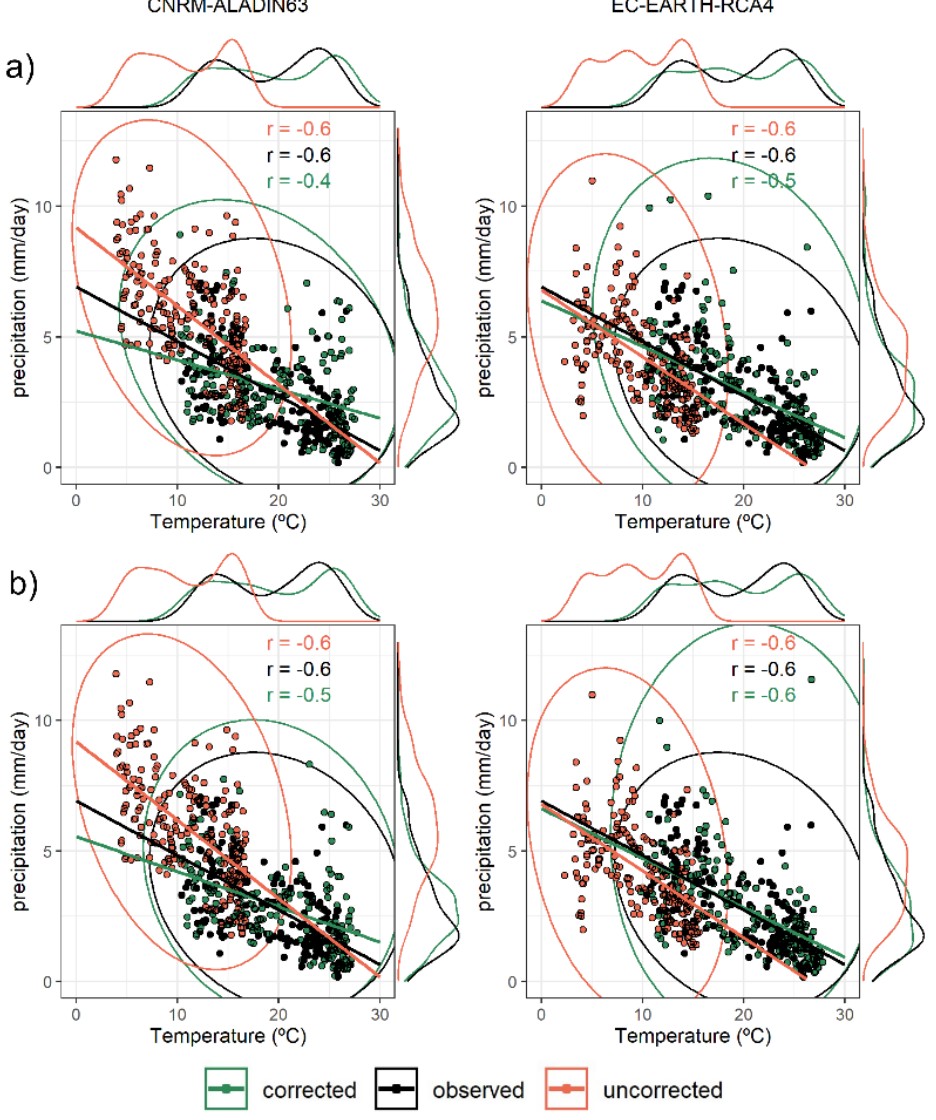

**Figure 7. Distribution of mean daily temperature versus mean daily precipitation from March to August (1981–2005) for the CANT region and for two RCM: CNRM-ALADIN63 and EC-EARTH-RCA4. UBC method (a); MBC method (b); Fitted lines, area of distribution and density distributions; green refers to the bias corrected model, black indicates the observed data, and red shows the raw/uncorrected model. Pearson correlation values (r) are also shown.**

The performance of the bias correction methods in reproducing the distribution of the extremely long dry spells is generally irregular and unable to reproduce the observed distribution in some cases, a phenomenon already pointed out by Maraun and Widmann (2018) and François et al. (2020). On analysing the ECDFs (Fig. 8) generated for the CANT region and for the two bias correction methods, the MBCn is seen to provide values of the $D$ statistic of the KS test closer to zero than the UBC method for all models except for the IPSL-RCA4. Furthermore, the ECDF of the MBCn is also observed to fit the observed distribution better than the ECDF of the uncorrected model, with the exception of the CNRM-ALADIN63 model. For the two bias correction methods and for the CANT region, only the CNRM-ALADIN63 model and the NORESM-HIRHAM5 model showed statistically significant KS test values at the 95 % confidence level; this indicates that only for these two cases, the duration of the BC extreme dry spells differs from the observed values. The NMED region presents very different results from those provided for the previous region. Neither of the bias correction methods can be seen to outperform the other. In both methods the bias correction fails to reproduce the observed ECDF (p-value < 0.05 in all models and bias correction methods). In the ECDFs there is clearly an underestimation of the length of the extreme dry spells both for the uncorrected model and for the model after correction by the two BC methods. Thus, we observe that although the results for the CANT region are quite accurate, there exists a high degree of uncertainty in the estimation of dry spells in the NMED dipole region, which implies that the results to be projected in subsequent analyses should be considered caution.

In the case of temperature extremes, both bias correction methods perform well at extreme percentile daily temperature p95 for the CANT and NMED dipole regions, with no apparent bias in performance (Fig. 9). On reaching more extreme values, such as the temperature extremes (p95) occurring within periods of extremely long dry spells (p95), which we call EM, the MBCn-corrected models are generally seen to reproduce these very extreme temperature values much better than the UBC-corrected ones. These differences are more noteworthy in the CANT region than in the NMED region. The patterns detected for these two regions are very similar to the remaining regions of the study area (Fig. S4).

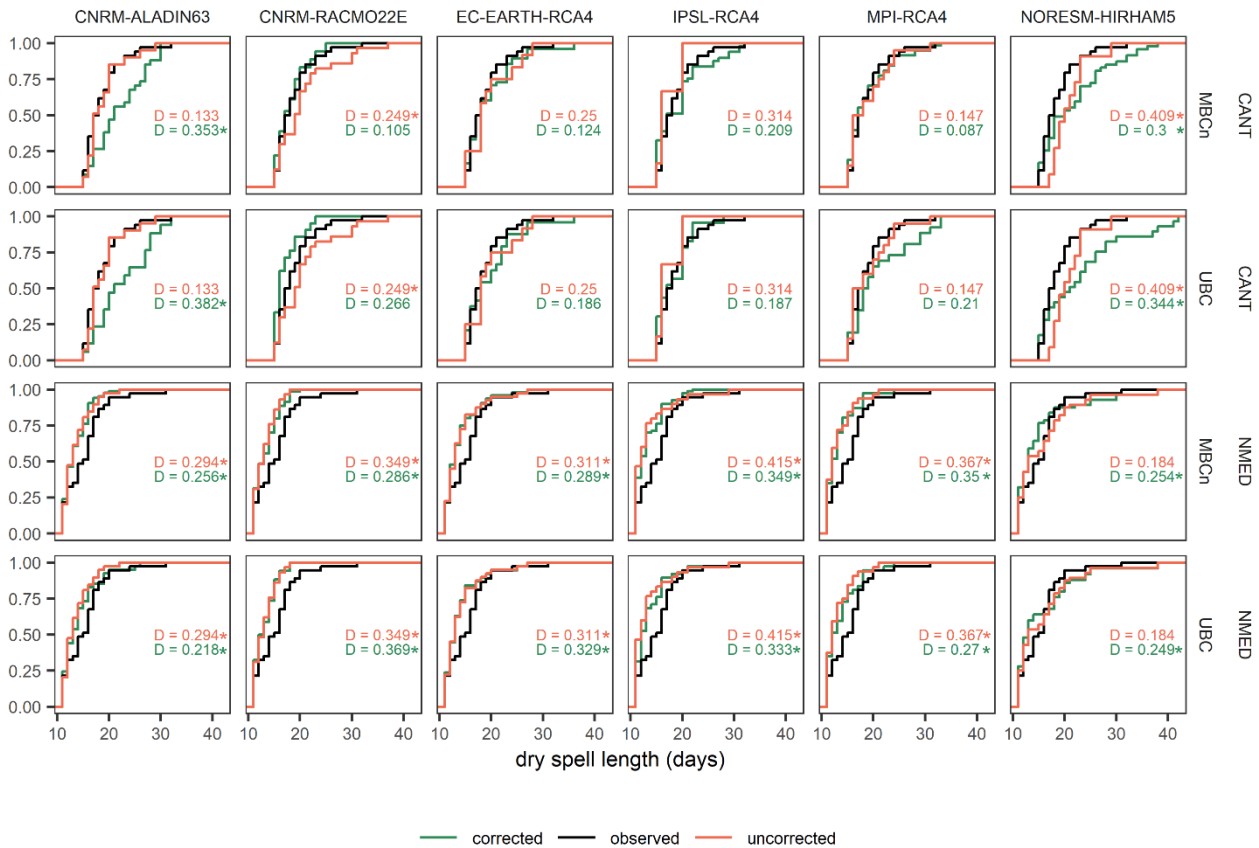

**Figure 8.** Empirical cumulative density function of the extreme dry spells length (p95) according to the CANT and NMED regions, UBC and MBCn bias correction methods, and the 6 historical RCM used in this study for the 1981-2005 period. The Kolmogorov-Smirnov statistic is annotated in each plot as D. The asterisk indicates that the null hypothesis of the KS test is rejected. Green shows the corrected distribution, red the uncorrected distribution and black indicates the observed distribution.

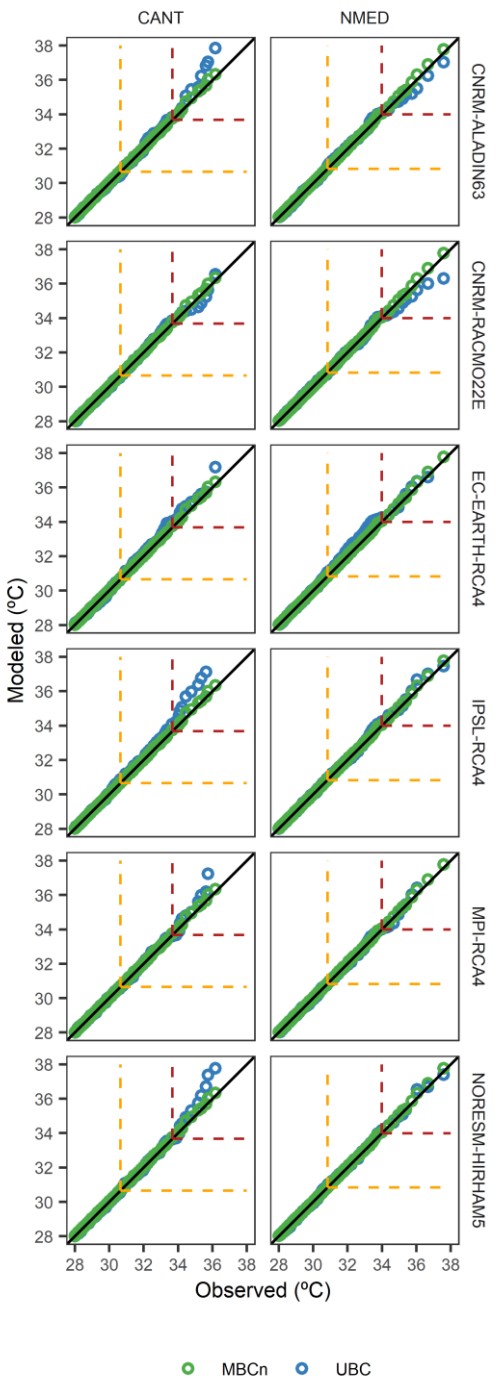

**Figure 9. Quantile–quantile plot of observed versus modeled values for temperatures in the CANT and NMED regions and all RCMs; green and blue circles indicate MBCn and the UBC method, respectively. The yellow dashed box at top right shows the 95th**

**percentile of observed daily temperature, while the dark red dashed box at top right shows the 95th percentile of observed daily**

**temperature during the occurrence of dry spells of extreme length (95$^{th}$ percentile).**

### 5. Future changes in the variables underlying the compound event.

After evaluating the bias-correction methods, in this section we present the results of the bias-corrected projections using the MBCn method. The regional projections showed an increase in the duration of D events (Fig. 10a); these were only abundant

in the case of the scenario of high greenhouse emissions (RCP8.5) and were consistent across all regions during spring. On the other hand, in summer substantial increases are only detected in the EATL, HIPY and NMED regions, which are all located in the northern half of the Pyrenees. However, the scenario projected by the RCP4.5 contrasts greatly with the previous one. In this moderate RCP4.5 scenario, with the exception of the WCONT and ECONT continental regions, none showed any statistical significance in the duration of D events during spring. In summer, and under the latter scenario, no statistically

significant trend was detected towards an increase in D events in any of the regions.

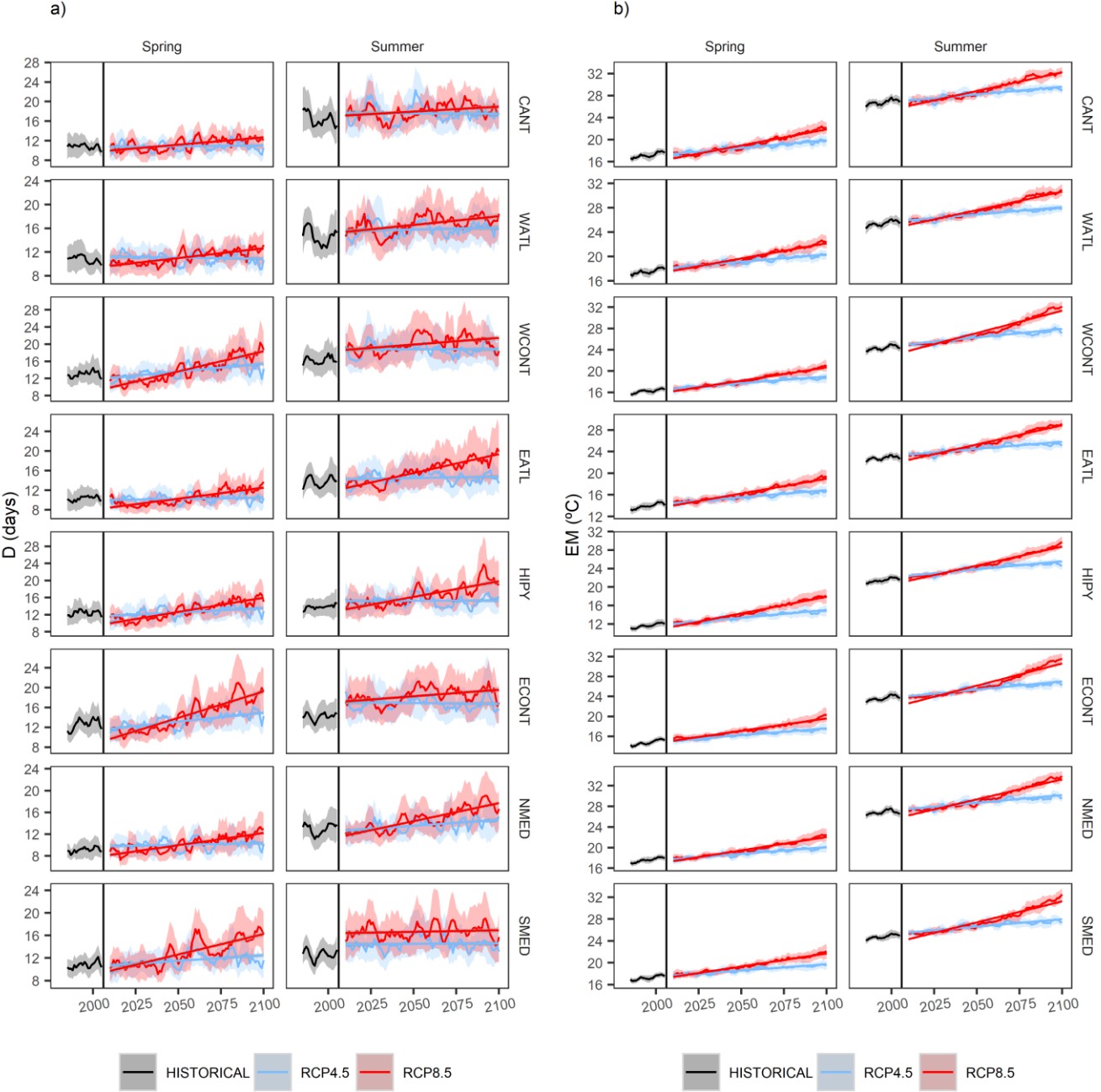

**Figure 10. Observed period and historical (1981-2005) and future (2006-2100) projections (5-year moving average) of the events of a) D and b) EM for an intermediate (RCP4.5) and extreme (RCP8.5) scenarios for all regions and for the spring and summer seasons. The curves show the average value of all models, while the shaded area indicates the standard deviation of the models for each year**

In the case of the hot extremes (EM), the previously detected increase was evident under both scenarios (Fig. 10b). However, special attention should be paid to the greater increase in EM in relation to M (Fig. 11).

Moreover, the rate of warming during the hot extremes was variable albeit more consistent in a high-emission scenario (RCP8.5) (Fig. 11). Interestingly, under this scenario and during the spring, the EM trend was above M throughout the study area, with particular incidence in the CANT, EATL, ECONT regions, about 0.10ºC per 10 years. In summer, the increase in EM was faster than that in M in the southern regions, especially in WCONT, ECONT and SMED, up to 0.15ºC per 10 years more rapidly than in M trends. In the intermediate scenario, there was greater equilibrium between the EM and M trends, but in all regions, there is a trend towards a faster increase in EM than in M.

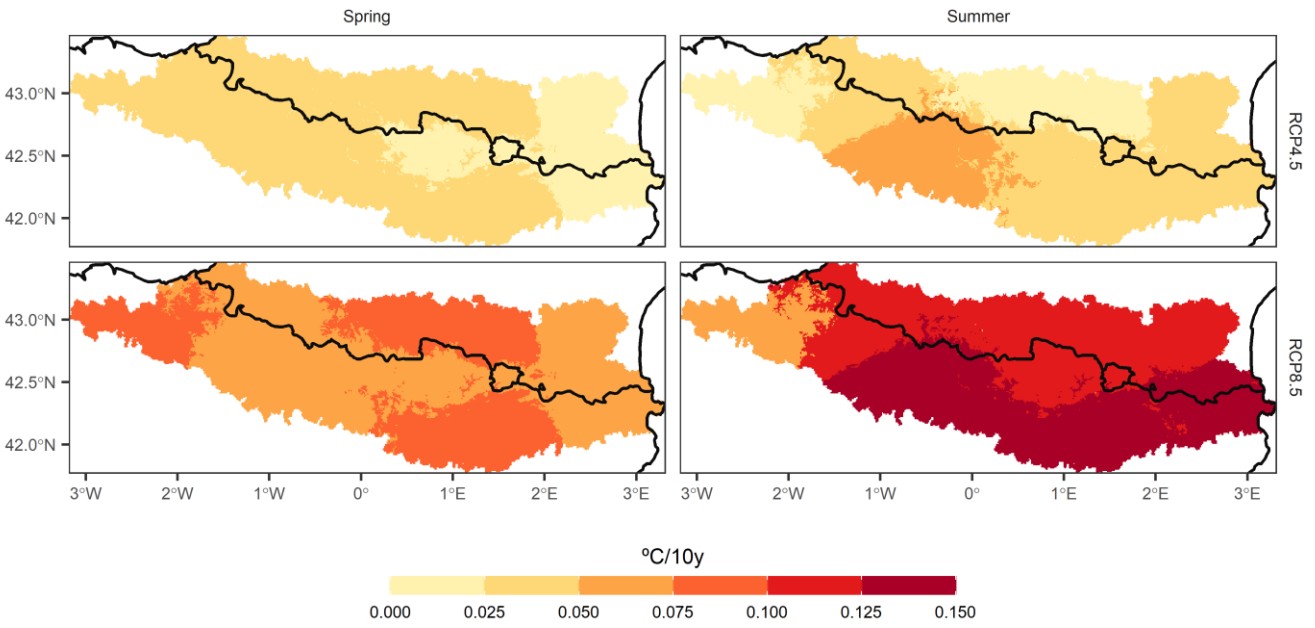

**Figure 11. Multi-model mean projected change between the trends of M and of EM variables. Positive values indicate a higher positive trend in EM values than in M values.**

It is of particular interest to analyze these D and EM events jointly to ascertain whether the compound risk of these two variables will be equally distributed or whether each of the two variables will have a different weighting in the joint event. This evaluation is shown in Figure 12 for the CANT and NMED dipole regions, where the multivariate coordinates of the anomalies of events D and EM are shown; these are divided into three periods: (2016-2035, 2046-2065 and 2081-2100), which are consistent with the periods selected in the IPCC Fifth Assessment Report (Stocker, 2014), for both seasons and both emission scenarios. On the one hand, the displacement along the Y axis enabled the increase in the duration dimension (D) in the compound event to be evaluated. On the other hand, the displacement along the X axis indicated an increase in the thermal anomaly and consequently, greater risk posed by the magnitude dimension (EM). In the case of the CANT region, the average

value of the bivariate distribution of each spring and summer period projected by the RCP4.5 clearly indicated that the increase in the compound risk was caused by an increase in extreme magnitude (EM), i.e. by the thermal increase, as opposed to an increase in the duration of such events (D). The same assessment can be extrapolated to the NMED region for the spring in a RCP4.5 scenario. A very similar pattern is observed in the case of summer for this intermediate scenario. In the RCP8.5 scenario, a very considerable increase in risk was perceived as a result of the increased weight of the magnitude, especially in the last two periods in both seasons and regions. The increase in the D dimension continued to be very weak for the CANT region, regardless of the season analyzed. On the other hand, in the NMED region, there was a remarkable increase in dimension D, which rose by an average of 5 days (summer, 2081-2100) with respect to the historical average (1981-2005). In this case, we detected that a statistically significant increase (p-value < 0.05) in the compound risk occurred in both dimensions (up to 6ºC in summer), thus implying a much higher risk than in the CANT dipole region.

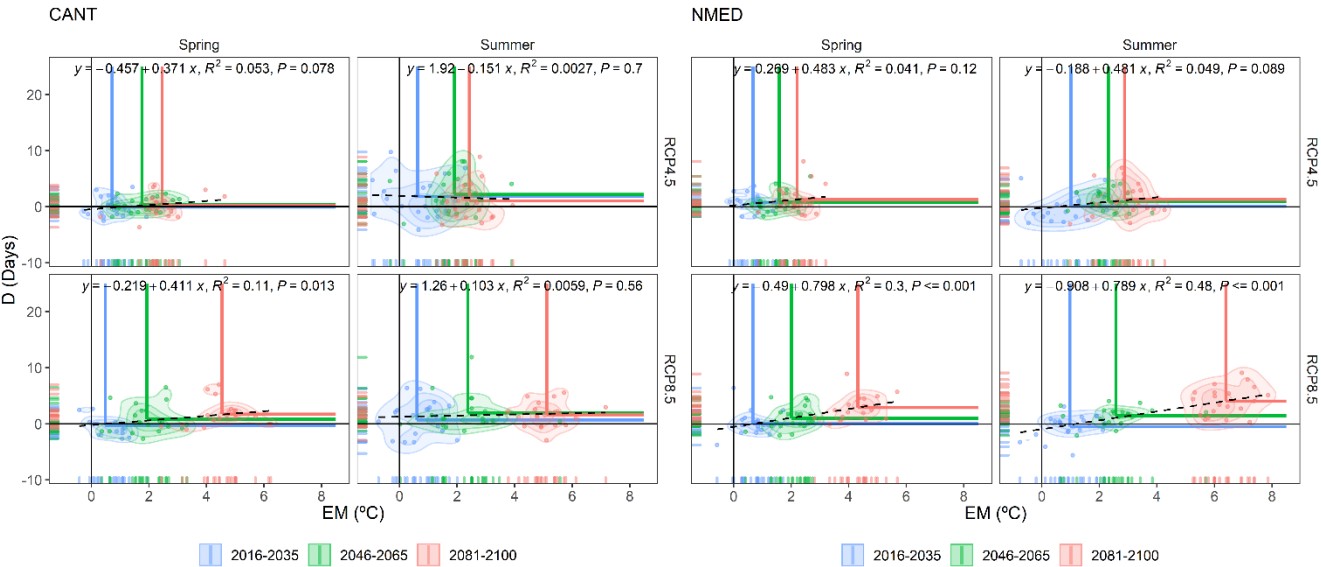

**Figure 12. Bivariate probability density functions of D and EM anomalies for the three future periods (2016-2035, 2046-2065 and 2081-2100) and for the two emission and seasonal scenarios with respect to the historical period (1981-2005) for the CANT and NMED regions. Each point in the scatter plot represents the multi-model annual mean of D and EM in a given year. The intersections of the blue, green and red horizontal and vertical lines indicate the mean anomaly value of the bivariate distribution for each period. The linear fit regression was computed using the annual mean anomalies of EM and D for the 2016-2100 period. Each plot possesses a regression equation and its statistical significance (*P*, p-value). The figure is generated using the ensemble of all RCMs.**

With regard to the other regions, several patterns are observed across the study area. i) For RCP4.5, in all regions and in both seasons, there is a noteworthy increase in the EM dimension, while no changes occur in the D dimension. ii) However, in the RCP8.5 scenario for spring, all regions show an increase in the compound risk as a result of an increase in the duration of both D and EM events. iii) For this extreme scenario in summer, only the EATL, HIPY and NMED regions, and to a lesser extent,

the WATL region, all located in the northern half of the Pyrenees, show an increase in both dimensions (D and EM). The rest only exhibit an increase in the thermal dimension (EM), see Figs. S5-S9, for details.

Of particular interest is the HIPY region, which presents the highest average elevation in the study area, with several glaciers and a multitude of snow-capped mountains. Precisely, this region in an RCP8.5 scenario will present a marked statistically significant increase (p-value < 0.05) in both dimensions (Fig. 13). This will occur gradually both in spring and summer during this century.

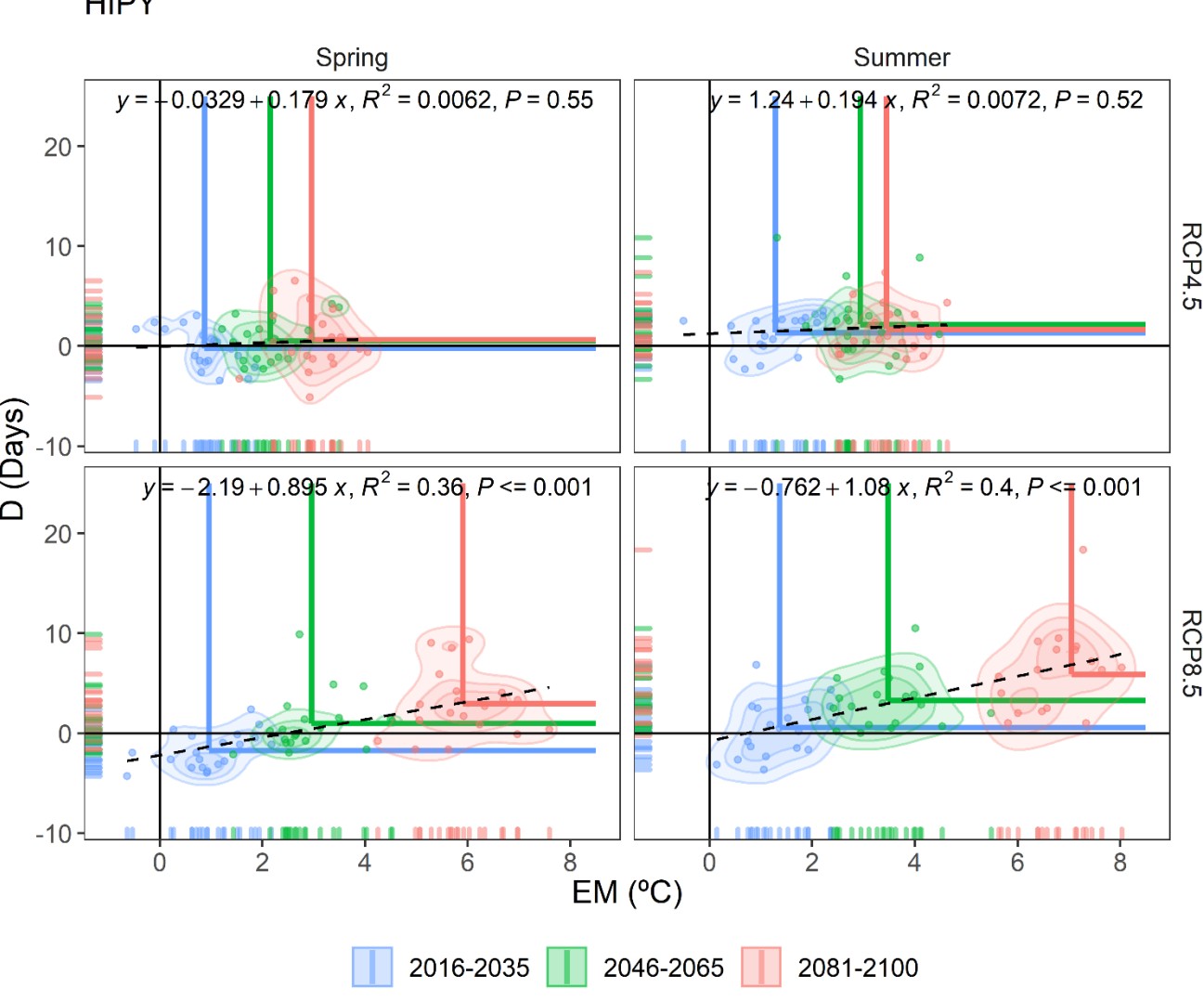

**Figure 13. Same as Fig. 12 but for HIPY region.**

These results were summarized in Figure 14, which shows the future patterns of Dry-Hot compound events according to the D and EM variables for the whole Pyrenees area. Each season and emission scenario present a different pattern, summarized below:

- **Spring | RCP4.5**: Increase in one-dimensional compound risk based on (extreme) magnitude.

- **Summer | RCP4.5**: Increase in the one-dimensional compound risk based mainly on (extreme) magnitude. Although
a greater increase in EM is observed in all regions (see Fig. 12 and Fig. 13, for CANT, NMED and HIPY regions), no increase in the second dimension (extreme length of dry spells) is detected.

- **Spring | RCP8.5**: Increased risk resulting from a two-dimensional component in all regions of the Pyrenees. The increase in extreme magnitude is slightly greater in the Mediterranean (NMED and SMED) and continental regions

(ECONT and WCONT). A sharp increase is observed in the second dimension (D) in all regions, except for the westernmost regions (CANT and WATL) (moderate increase).

- **Summer | RCP8.5**: Increase in two-dimensional compound risk in the northern façade of the Pyrenees (NMED, EATL, HIPY and, to a lesser degree, WATL). The increase in the other regions mainly refers to the EM dimension. The increase in EM in the final period in all regions is the highest of the four patterns described.

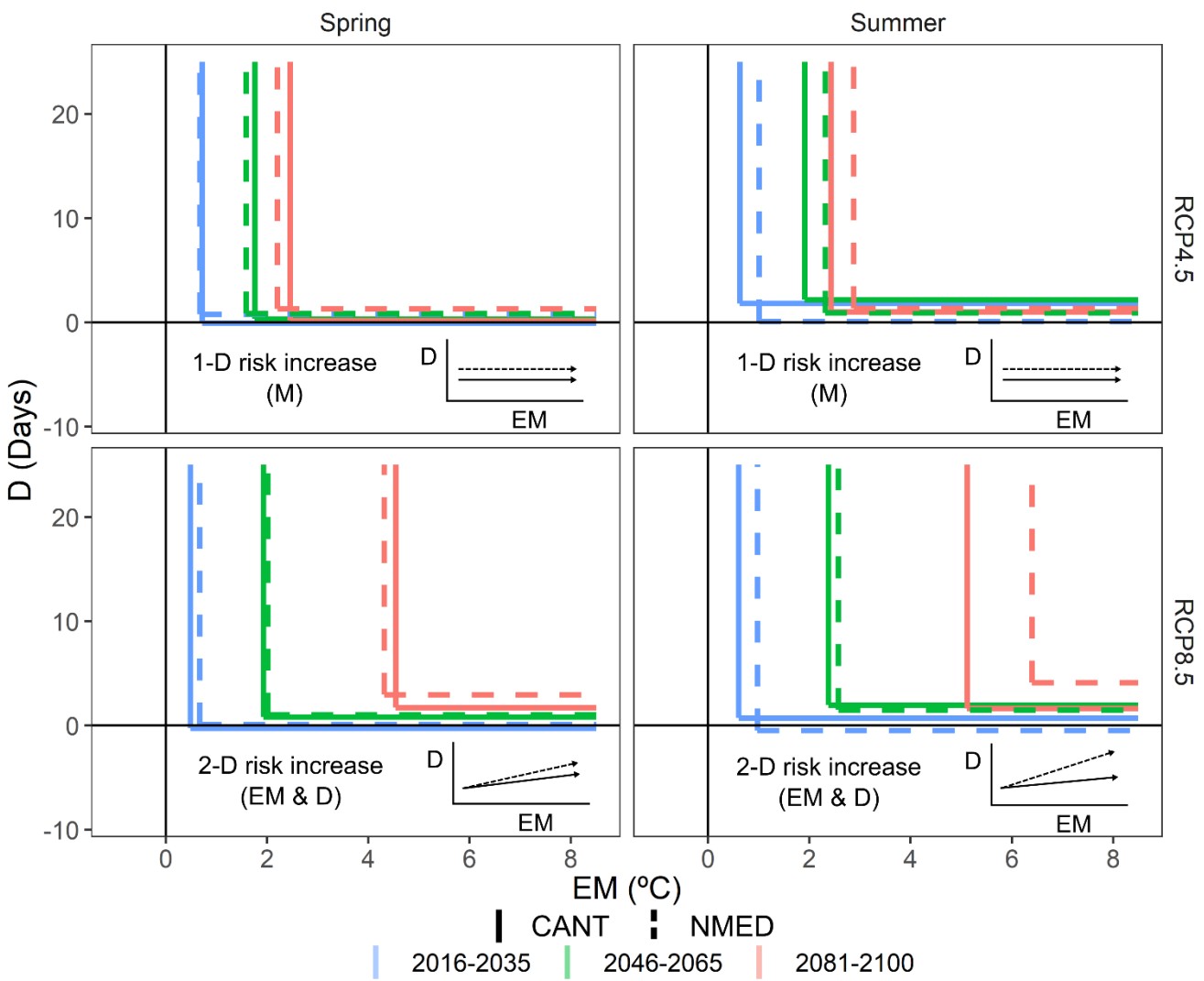

**Figure 14. Diagram of the drivers of the three future periods (2016-2035, 2046-2065 and 2081-2100) of the compound event for D and EM in the CANT (solid line) and NMED (dashed line) regions, which acted as a dipole in the study area.**

The remaining regions shown in the supplementary document were at the intermediate stages of those described herein (Figs. S5-S9). However, the WATL region, which is adjacent to the CANT region, was influenced by both climates (Atlantic and Mediterranean) and therefore did not reflect a similar behavior pattern to that of the CANT region (see Fig. 1 to verify the heterogeneity of this region).

## 6. Discussion

The use of bias correction methods to correct the distribution of dry spells indicated the inability to robustly resolve these types of biases. The bias correction only resolves the bias resulting from the drizzle effect but not the biases resulting from topographic issues or the underestimation of the persistence of anticyclonic conditions (Maraun et al., 2017). For future studies, as suggested by Maraun et al. (2021), before applying bias-correction methods, a prior evaluation of the performance of GCMs in simulating the persistence of dry events is needed mainly to transfer the minimum bias in the temporal dependence when applying the bias correction.

On the other hand, the results of the present research reveal that up to the present there has been a general increase in the compound risk of Dry-Hot events due to an increase in the thermal component; thus, the duration dimension is excluded, as pointed out in various recent studies (Hao et al., 2018; Manning et al., 2019). A significant finding of our study refers to a significant increase throughout the Pyrenees, and to the compound risk in relation both to the magnitude dimension (extreme temperature) and to the duration dimension (duration of extreme dry event), for spring under the RCP8.5 scenario. A sharp increase was also detected in both dimensions for the northernmost regions of the study area during summer under the RCP8.5 scenario. Therefore, it was estimated that in the future the compound event will exhibit a more balanced distribution between the two dimensions, with the D dimension gaining prominence. Polade et al., (2014) showed that the areas in which a greater increase in dry days is expected under a RCP8.5 scenario are the western Mediterranean and the eastern Atlantic, at approximately latitudes 35ºN and 55ºN, in accordance with the findings of the present study.

Nonetheless, there are conflicting opinions regarding whether the observed warming is inducing an increase in the length of dry spells, as noted by Ye and Fetzer (2019) in Russia, or whether, as observed by Trenberth et al., (2014), the warming does not prolong the dry event, but that the warming itself may augment the intensity of the episode due to the effect of thermal magnitude. The authors consider that the observed warming has not caused longer-lasting droughts in the area of the Pyrenees, which does not correspond with the findings of Ye and Fetzer (2019). Furthermore, even under an intermediate emissions scenario (RCP4.5), there is no evidence of increasingly longer dry spells, despite a significant rise in temperature. Consequently, the authors do not support the idea that a thermal increase is directly related to longer dry periods. This greater duration of dry spells in the most extreme emission scenarios may be due to northward shifting of the subtropical anticyclone belt (Gillett and Stott 2009) as a consequence of the expansion of the Hadley cell in response to global warming (Lu et al., 2007). There is a need for further research in order to understand the future role of the subtropical anticyclone belt and

variations therein, and how these affect increases or decreases in compound events, considering the fact that subtropical ridges are the main drivers of these extreme events (Fig. 5).

**7. Conclusions**

The risk posed by the simultaneous occurrence of extremely dry and hot events is analyzed for the first time in the Pyrenees, which is a very fragile area as a result of its altitude and transition latitude between the temperate and subtropical climates. Moreover, 59% of its area is covered by forests, which will become susceptible to severe wildfires if climatic conditions are unfavorable during the following years. We extracted the following main findings from the present study:

- The results for the observed period (1981-2015) showed a generalized increase in the thermal extremes (EM) within the extreme dry spells (D), with no increase in the duration of these spells. This showed that to date the compound risk has only augmented in one dimension: Extreme magnitude (EM) and by default, magnitude (M).

- As regards the results obtained from future projections, it is essential that an intermediate emission scenario (RCP4.5) not be exceeded, as this serves to prevent the D dimension (duration) of such events from increasing. The compound
risk keeps rising, but only because of the even more pronounced thermal increase.

- In a high-emission scenario (RCP8.5), the increased risk of the compound event would be a consequence of an increase in both the extreme magnitude (EM) and the duration (D) dimensions. In addition, and within this context, the thermal increase in extremely hot days (EM) during the dry event (D) is greater than the thermal increase in the set of days (M) comprising the dry event (D).

Finally, and by way of a general conclusion, the present study reveals a potential increase in environmental risks in the Pyrenees (fires, crop yield losses, effects on biodiversity, water resources, etc.) resulting from more frequent compound events involving long dry periods and extreme temperatures. The high-altitude and northernmost regions could be affected to a greater extent, regardless of the season (spring or summer) during the high-emissions scenario. There also exists a need to study whether other natural hazards such as wildfires are observed during such extremely hot intervals within these very long dry periods, in
order to prepare this area to tackle large wildfires.

**Acknowledgements**

The present research was conducted within the framework of the Climatology Group of the University of Barcelona (2017 SGR 1362, Catalan Government), and the CLICES project (CGL2017-83866-C3-2-R, AEI/FEDER, UE). We wish to thank Dr. Cuadrat and Dr. Serrano-Notivoli for the CLIMPY project database. M.L-C was awarded a pre-doctoral FPU Grant
(FPU2017/02166) from the Spanish Ministry of Science, Innovation and Universities. The authors declare no conflict of interest.

## Data availability

CLIMPY database is available from: https://zenodo.org/record/3611127#.X_NxyRaCFPY. EURO-CORDEX projections are available from https://cds.climate.copernicus.eu/cdsapp#!/dataset/projections-cordex-domains-single-levels?tab=overview.


## Author contribution

M.L-C and J.A L-B: Conceptualization. ML-C: Methodology, Data curation, Writing-Original draft preparation. JA.L-B: Supervision

## Competing Interests

The authors declare that they have no conflict of interest.

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
