# Peer review of "Assessing internal changes in the future structure of Dry-Hot compound events. The case of the Pyrenees"

_Natural Hazards and Earth System Sciences, 2021_

## Referee Comment (RC1)

This paper assesses changes in the duration of dry spells and temperatures during spells over the Pyranees for an observation period (1981-2015) and in future projections from an ensemble of regional climate models. The authors find a significant increase in temperatures during dry spells over the observation period and in projections from future climate simulations, but little change in the duration of dry spells. The paper is generally well presented but there remains many aspects that need clarification in their methods and in the presentation of their results. For instance, it is unclear in many figures what exactly has been done to produce the specified result. There is also a major issue with the use of bias correction, which is unjustified in this case, particularly with respect to bias correcting the duration of dry spells. These biases likely result from biases in the persistence of large-scale anti-cyclones which cannot be corrected simply via quantile mapping. Therefore, I unfortunately cannot recommend the paper for publication in its current state. However, I feel this research is very relevant and could be brought to publication standard. Specifically, through better discussion of the model biases, the potential sources of such biases, and the how such climate model limitations impact the confidence we can have in the future projections of such events. The reasoning behind my decision is explained in more detail below.

Comments (**P: Page, L: Line Number**):

- P3 L84: The analysis by Jacob et al. (2014) does not include any validation, only an analysis of future changes in a range of metrics in these simulations.

- Regionalisation:
  - P4 L97-105: The authors suggest that there is no variability throughout the assessed region when long dry spells occur as there is an "identical synoptic behaviour pattern throughout the region". What is the motivation is for using this regionalisation approach for the analysis of long dry spells?
  - P4 L112: what is meant by iterations in this case, and how does this ensure a robust regionalisation? Please explain more precisely.
  - P4 L113-114: What variance is it explaining? Daily temperature and precipitation?
  - P4 L113-114: How is the total explained variance calculated? What output is given from the k-means algorithm to do this? Please be more precise.

- Event definition:
  - P6 L139-140: What is meant by annually? Do you mean that you extract only one event per year?
  - P6 L140-142: Does this mean that there are multiple EM events in one dry spell? If so, do you consider all of these 'EM events' as independent events such that there would be more EM events than M or D events by definition? Please explain more precisely.

- Bias correction
  - I'm not convinced that bias correction via quantile mapping is appropriate here. It is a simple method that is used to make simple corrections to climate model output. The method only adjusts each quantile of the RCM distribution to the corresponding quantile of the observed distribution, and so it is trivial that the bias corrected distribution will be similar to the observed distribution, as is shown in Figure 6. See Maraun et al. (2017) who consider an extreme example of quantile mapping in which

the distribution of temperature from the Pacific Ocean to precipitation in central Europe. If using quantile mapping, it should be clear what is driving the bias in a given variable.

- o Quantile mapping is particularly inappropriate in the case of duration. The biases seen in duration (Figure 6) are derived from the lack of persistence of dry days in the underlying precipitation time series. This itself is driven by a lack of persistence in large-scale drivers (e.g. persistent anti-cyclones). A simple bias correction via quantile mapping of the Duration distribution cannot correct biases in the large-scale circulation and will result in fictitious events in the bias corrected distribution.
- o Furthermore, quantile mapping in this case will just hide significant and relevant uncertainties in these climate model simulations. For instance, if the models cannot represent the persistence of long dry spells, then we cannot know with any confidence how such events might change in the future. This is a reality we are faced with in the community which cannot be simply fixed via quantile mapping.
- o The uncertainty is hidden in the results obtained from the bias corrected distributions, and I would not be confident in the robustness of the results. I think it would be more informative for the authors to present the relevant biases of these models and discuss the implications of such biases for the future projections. This could help as feedback to model developers in order to improve these climate model biases.

- Figure 3: Have you taken the average of all events that exceed the local 90$^{th}$ percentile? Is this average sensitive to the occurrence of single events? The figure seems a little noisy in places which might not be expected for metrics of such large-scale events. I'd imagine looking at the 95$^{th}$ percentile would be more robust.

- Figure 4: The average of EM is larger than that of M by construction of the analysis, it is a trivial result. You are comparing the unconditional distribution of M with the distribution of EM which is a conditional distribution of temperature given that it exceeds the 90$^{th}$ percentile. EM is different from M because you impose a threshold on temperature. Maybe I have missed the point but I do not see the relevance of this figure, please explain the significance of this result. Is it simply that the average temperature of dry spells with temperatures above the 90$^{th}$ percentile are warmer than dry spells where no threshold is imposed?

- Figure 5: What were these trends calculated for? Are all events considered or just the annual maximum? If it's the former, how is the resulting slope interpreted given that there will be a different number of events each year?

- Figure 6: What are the biases calculated between? The mean of the distributions or some other metric? Please specify.

- Figure 8: What is the 7-year moving average taken of? From all events in the 7-year period? Please specify.

- Figures 10, 11, 12: This is a nice of visualising the change in the bivariate distribution. However, there are a number of aspects that need clarification:
  - o Is this figure for one model only? Or do you pool the events from all models into one distribution?

- How do you compute the linear regression shown in each panel? Specifically, what values are used to construct it?
- From your definition of EM, you would obtain multiple values of EM per event. What do you plot against Duration in these figures? Is it one EM value per event? Or is each EM value considered such that the same event would be repeated multiple times in the scatter plot?

- P. 22 L337-338: It is mentioned that there is no change in duration, but the NMED and SMED regions show an increase in mean Duration for RCP8.5 in Spring and Summer (Figures 10 and 11), and it seems there are more very long duration events also from counting the number of dots in the scatter plot.

---

## Author Response (AR1)

**REVIEWER 1**

This paper assesses changes in the duration of dry spells and temperatures during spells over the Pyrenees for a given observation period (1981-2015) and in future projections from an ensemble of regional climate models. The authors find a significant increase in temperatures during dry spells over the observation period and in projections from future climate simulations, but little change in the duration of dry spells. The paper is generally well presented but there remains many aspects that needclarification in their methods and in the presentation of their results. For instance, it is unclear in manyfigures what exactly has been done to produce the specified result. There is also a major issue with the use of bias correction, which is unjustified in this case, particularly with respect to bias correctingthe duration of dry spells. These biases likely result from biases in the persistence of large-scale anti- cyclones which cannot be corrected simply via quantile mapping. Therefore, I unfortunately cannot recommend the paper for publication in its current state. However, I feel this research is very relevant and could be brought to publication standard. Specifically, through better discussion of the model biases, the potential sources of such biases, and the how such climate model limitations impact the confidence we can have in the future projections of such events. The reasoning behind my decision isexplained in more detail below.

*"We gratefully acknowledge the reviewer's comments and the revision of our manuscript. The article has been revised in accordance with the referee's comments and suggestions, which are addressed below. We have paid particular attention to improving the presentation of the results because the principal weakness of the paper detected by the referee involves a degree of uncertainty in the projected outcome as a result of bias correction methods. We believe that his comments have helped to make a significant improvement in the manuscript. Our answers appear in italics and in quotation marks"*

Comments (**P: Page, L: Line Number**):

- P3 L84: The analysis by Jacob et al. (2014) does not include any validation, only an analysis offuture changes in a range of metrics in these simulations.

*"We agree with the reviewer; we have eliminated the term validation from this sentence. See L93-94"*.

- Regionalisation:
    - P4 L97-105: The authors suggest that there is no variability throughout the assessed region when long dry spells occur as there is an "identical synoptic behaviour pattern throughout the region". What is the motivation is for using this regionalisation approach for the analysis of long dry spells?

        *"We agree with the reviewer. This paragraph was somewhat confusing. We have added a new paragraph between L113-117 to explain our reason for dividing the Pyrenees into a few basic regions. In short, although Hot-Dry events are derived from synoptic situations arising in subtropical ridges (Fig. 5), this kind of situation exerts a greater impact upon the southern area of the Pyrenees than on the northern sector, thus producing different intensities of Hot-Dry compound events throughout the study area. It is therefore of interest to divide the Pyrenees into a number of basic regions that can differ in their behavior patterns."*

    - P4 L112: what is meant by iterations in this case, and how does this ensure a robust regionalisation? Please explain more precisely.

        *"We extended the explanation in L124-126 to make it more comprehensible"*

    - P4 L113-114: What variance is it explaining? Daily temperature and precipitation?

        *"The variance explained is approximately 48 % (See L128-134). This percentage appears to be low, but this is because we are selecting few regions in a study area presenting a high topographic complexity. In addition, using two variables (temperature and precipitation, previously scaled), provides greater variability to this area. Selecting a higher percentage of variance would involve excessive division of the study area."*

o P4 L113-114: How is the total explained variance calculated? What output is given from the k-means algorithm to do this? Please be more precise.
*"We added to the text a precise explanation about how to compute the explained variance (%). See L128-134."*

- Event definition:
  o P6 L139-140: What is meant by annually? Do you mean that you extract only one event per year?
  *"In each year we selected the consecutive days comprising dry spells with a duration greater than the 95ᵗʰ percentile of each year. By way of an example, in the grid cell i,j based on the 95ᵗʰ percentile of dry spell lengths of 1981, there were 2 dry spells accounting for 15 and 20 days, respectively. This process is repeated for each year. We have rephrased it in the manuscript. See L.159-160."*
  *"We detected an error in Figs. 3 and 4, which were not calculated annually, but rather for the whole study period. For this reason, some dry spells lasted longer than 100 days in Fig. 3 (when summer and spring separately only score approximately 90 days)."*

  o P6 L140-142: Does this mean that there are multiple EM events in one dry spell? If so, do you consider all of these 'EM events' as independent events such that there would be more EM events than M or D events by definition? Please explain more precisely.
  *"EM events depend on the occurrence of D events. Throughout the text we emphasize that M and EM events only occur during a D occurrence. M is only the temperature of each day of a D (extreme dry spell), and EM is the p95 temperature occurring during an extreme dry spell. Extreme temperatures outside these dry spells are not considered." (See L162-163)*

- Bias correction
*"We are truly grateful for your extensive comments on the bias correction performance. We have now addressed all your questions at the end of this section"*

  o I'm not convinced that bias correction via quantile mapping is appropriate here. It is a simple method that is used to make simple corrections to climate model output. The method only adjusts each quantile of the RCM distribution to the corresponding quantile of the observed distribution, and so it is trivial that the bias corrected distribution will be similar to the observed distribution, as is shown in Figure 6. See Maraun et al. (2017) who consider an extreme example of quantile mapping in which the distribution of temperature from the Pacific Ocean to precipitation in central Europe. If using quantile mapping, it should be clear what is driving the bias in a given variable.
  o Quantile mapping is particularly inappropriate in the case of duration. The biases seen in duration (Figure 6) are derived from the lack of persistence of dry days in the underlying precipitation time series. This itself is driven by a lack of persistence in large-scale drivers (e.g. persistent anti-cyclones). A simple bias correction via quantile mapping of the Duration distribution cannot correct biases in the large-scale circulation and will result in fictitious events in the bias corrected distribution.
  o Furthermore, quantile mapping in this case will just hide significant and relevant uncertainties in these climate model simulations. For instance, if the models cannot represent the persistence of long dry spells, then we cannot know with any confidence how such events might change in the future. This is a reality we are faced with in the community which cannot be simply fixed via quantile mapping.
  o The uncertainty is hidden in the results obtained from the bias corrected distributions, and I would not be confident in the robustness of the results. I think it would be more informative for the authors to present the relevant biases of these models and discuss the implications of such biases for the future projections. This could help as feedback to

model developers in order to improve these climate model biases.

*"The authors have made structural changes in the performance of the bias correction to guarantee more correct and transparent results in terms of the uncertainty of this type of correction in climate simulations. Below we summarize the changes made in the methodological aspect (integral restructuring of section 2.4) and to the assessment of the results of the bias correction methods (section 4):"*

- *"We first applied empirical quantile mapping (EQM) to the original temperature and precipitation data, rather than directly to D, M and EM. If the correction is performed for D, M and EM, the physical connection between the original precipitation and temperature can be omitted (i.e. the physical mechanisms are lost)."*

- *"Subsequently, we incorporated a multivariate BC method, the Multivariate Bias Correction with N-dimensional probability density function transform (MBCn), proposed by Cannon (2018). This method enabled us to maintain the structural dependence between temperature and precipitation (see Fig. 7, section 4), which is relevant when working with compound events. Throughout section 4, the performance of MBCn is compared to EQM."*

- *"To analyse the uncertainty in the estimation of dry spells in the climate models and in the subsequent correction, the D-statistic of the Kolmogorov-Smirnov test (L217-222) was used. Section 4 shows that the results are very irregular, which for the CANT region are acceptable, since the distribution simulated by the BC methods is close to the observed one; but this is not the case for the NMED region, where the distribution of the dry spells simulated and corrected by the two BC methods is clearly different according to the KS test. See Fig. 8."*

- *"Finally, the performance of the two BC methods in modelling daily temperature has also been analysed, with emphasis on the daily extreme values above p95 of Tx for each year (spring and summer), as well as on the daily extreme values of TX for each year within a dry spell. Section 4 explains that the EQM (denoted as UBC in the text (Univariate Bias Correction)) performs quite well, but when reaching the most extreme temperature values, those occurring within a dry spell, the MBCn is more accurate (Fig. 9 and Fig. S4)."*

- *"All these results suggest the need to employ MBCn when correcting future projections. This has led to some changes in section 5."*

- Figure 3: Have you taken the average of all events that exceed the local $90^{th}$ percentile? Is this average sensitive to the occurrence of single events? The figure seems a little noisy in places which might not be expected for metrics of such large-scale events. I'd imagine looking at the $95^{th}$ percentile would be more robust.

*"We also detected an error in Figs. 3 and 4, which were not calculated annually, but rather for the whole study period. Consequently, some dry spells lasted over 100 days in Fig. 3 (when summer and spring separately only have 90 days approximately). We also took into account the reviewer's suggestion and we applied the $95^{th}$ percentile throughout the study"*

- Figure 4: The average of EM is larger than that of M by construction of the analysis, it is a trivial result. You are comparing the unconditional distribution of M with the distribution of EM which is a conditional distribution of temperature given that it exceeds the $90^{th}$ percentile. EM is different from M because you impose a threshold on temperature. Maybe I have missed the

point but I do not see the relevance of this figure, please explain the significance of this result. Is it simply that the average temperature of dry spells with temperatures above the 90ᵗʰ percentile are warmer than dry spells where no threshold is imposed?

*"The authors consider that it is vital to conserve this figure in the manuscript, because it indicates that a dry spell "per se" implies slightly positive temperature anomalies, but during this drought period, maximum temperatures can reach extreme values, which can cause large wildfires, for example. We explain this in L167-168. In addition, we added to the manuscript a new figure (Fig. 5) which attempts to account for the physical mechanisms at play in these compound events. In this figure, a flash heat anomaly is seen to occur over the Pyrenees during the EM events."*

- Figure 5: What were these trends calculated for? Are all events considered or just the annual maximum? If it's the former, how is the resulting slope interpreted given that there will be a different number of events each year?

  *"Now Fig. 6, the figure caption is rephrased and is now more coherent."*

- Figure 6: What are the biases calculated between? The mean of the distributions or some other metric? Please specify.

  *"This figure was removed. Please note the new figures provided in this new version of the manuscript (Fig. 7, Fig. 8 and Fig. 9)"*

- Figure 8: What is the 7-year moving average taken of? From all events in the 7-year period? Please specify.

  *"Now, Fig. 10. The authors considered that it is already specified clearly in the caption of this figure. We changed from a 7-year moving average to a 5-year moving average to be more consistent with most of the studies in this material"*

- Figures 10, 11, 12: This is a nice of visualising the change in the bivariate distribution. However, there are a number of aspects that need clarification:
  - Is this figure for one model only? Or do you pool the events from all models into one distribution?

    *"We provide the results for the ensemble of all the models used. We have specified this in the caption of figure 12."*
  - How do you compute the linear regression shown in each panel? Specifically, what values are used to construct it?

    *"We have specified its construction in the caption of Fig. 12."*

  - From your definition of EM, you would obtain multiple values of EM per event. What do you plot against Duration in these figures? Is it one EM value per event? Or is each EM value considered such that the same event would be repeated multiple times in the scatter plot?

    *"The linear fit regression was computed using the annual mean anomalies of EM and D for the 2006-2100 period. Therefore, each year has a unique value of D and EM, which can serve to visualize the scatterplot"*

- P. 22 L337-338: It is mentioned that there is no change in duration, but the NMED and SMED regions show an increase in mean Duration for RCP8.5 in Spring and Summer (Figures 10 and 11), and it seems there are more very long duration events also from counting the number of dots in the scatter plot.

*"We have rewritten the text from L377 to L403. Further explanation of Figures 12, 13 and 14 is now provided."*

**REVIEWER 2**

Thank you for the opportunity to review the manuscript "Assessing internal changes in the future structure of Dry-Hot compound events. The case of the Pyrenees". The manuscript analyses a relevant topic and of wide interest in the scientific community related to compound events. In this work, the authors present a novel compound analysis of concurrent extreme dry spells and extreme hot temperature events in spring (MAM) and summer (JJA) on the Pyrenees, for the present and future scenarios. The proposed definition of the Dry-Hot events considers the length of extreme dry spells and the maximum and extreme maximum temperatures during the dry spells. The results point that present increases in compound Dry-Hot events are mainly attributed to increasing extreme temperatures, while future increases of the compound event will be likely associated to increases in both dry spells duration and extreme temperatures. Overall, the manuscript is well-conceived and organized, and the findings would be worth of publishing in NHESS. Nevertheless, I have some comments and suggestions detailed below that the authors may consider clarifying in the manuscript.

*"We gratefully acknowledge the reviewer's comments and the revision of our manuscript. The article has been revised in accordance with the referee's comments and suggestions, which are addressed below. Our responses appear in italic and quotation marks"*

**Specific comments**

*Introduction*

- Line 29: The authors may consider to use 'compound manner' instead of 'composite manner' to avoid confusion.

  *"Done"*

- Line 43: I recommend including a short introduction in one sentence to the definition proposed by Manning et al (2019) before this line.

  *"Done"*

*Data and methods*

- Line 76: I suggest the authors to provide some references about this: 'We focused on spring and summer, as spring can constitute the precursor of summer wildfires, and is a season prone to crop yield losses, etc.'

  *"Done"*

- Line 87: I suggest the authors to better explain why using the cell closest to the centroid of each region.

  *"Corrected. See L97-98"*

- Line 139-142:

  - I suggest the authors to better detail the estimation of the 90th percentile for D and EM events.

*"We have rephrased L159-160 to make clearer the event definition. In addition, we have changed the 90$^{th}$ percentile by 95$^{th}$ percentile to gain sensibility when detecting extreme dry spells and heat extremes."*

- Could Figure 2 be illustrated with a particular observational year?

  *"The authors considered that this scheme it is enough illustrative and allows the reader easily understand the meaning of each variable (D, M and EM)"*

- Would it be interesting to analyze the length of the EM events? (i.e. the number of consecutive extreme dry and hot days, in the addition of the value of Tx>90$^{th}$ during D events)

  *"The authors acknowledge the reviewer suggestion. However, the approach suggested for the reviewer could be biased due to the duration of D events. If in 1981 there are longer dry spells than in 2010, extreme temperature spells will be greater in the former case."*

*Results*

- Line 192: I suggest the authors to detail the calculation of the Tx anomalies, maybe in section 2.

  *"We added a short explanation in how we compute these anomalies. See L170-172."*

- Line 199: I think the authors meant Figure 3, like in the parenthesis in the end of the sentence.

  *"We have restructured the text of this part because we detected an error in Fig. 3 and Fig 4, which were not calculated annually, but for the whole study period. For this reason, there were dry spell lengths over 100 days in Fig. 3 (when summer and spring separately only have 90 days approximately)."*

- Line 255: I suggest the authors to justify the use of a 7 year-moving average, maybe in section 2.

  *"We have changed 7-yr by 5-yr, a more usual interval in this kind of applications."*

- Line 273-316:

  - I suggest the authors to describe in the section 2 the methods employed in the joint probability analysis of D and EM events.

    *"We have added more relevant information in the caption of Fig. 12, but the authors don't consider necessary to add more information to the section 2 because no additional methods are used to perform the Fig. 12, 13, 14."*

  - The physical interpretation of the mean value of the bivariate distribution in Figure 10 is the likelihood of average D occurring given that average EM occurs?

    *"This is the area of maximum point density (D and EM). More information was added to its caption to do this figure more comprehensive. Now it is Fig. 12."*

- o Maybe Figure 11 could be moved to Supplementary.

  *"We have changed Fig. 11 (now, Fig. 13), by the HIPY region bivariate distribution figure"*

- Line 283-284: "Nonetheless, in the case of summer for this same scenario, a small increase in the duration component was observed." I suggest adding to this sentence that the increase is higher from the first period to the second (2011-2040 to 2041-2070) than from the second to the third (2041-2070 to 2071-2100).

  *"There were structural changes in this part of this manuscript due to the application of a new method of bias correction MBCn, which implies a slightly different result."*

- Figure 10: Some information such as the regression line and the R-squared is not interpreted in the manuscript text.

  *"Done. See L369 or L385"*

- Line 301-316: I suggest specifying in parenthesis the Mediterranean and continental regions mentioned in the text, for example: Mediterranean regions (NMED and SMED).

  *"Done"*

*Discussion*

Line 327-329: 'A significant finding of our study indicates that there will be a significant increase in the future compound risk in relation both to the magnitude dimension (extreme temperature) and the duration dimension (duration of extreme dry event).' I suggest rephrasing indicating the regions, seasons and scenarios to which this finding applies.

*"We have rephrased these lines following the reviewer's suggestions. See L413-416."*

**Technical corrections**

- In general, figure captions can be improved, e.g.:

  - o Figure 4 – Indicate period as in Figure 3.

    *"Done"*

  - o Figure 6 – Explain better the horizontal interval in D points.

    *"Figure 6 was removed from the analysis due to the restructuration of section 4"*

  - o Figure 10 – Explain the black dashed line, the black numbers over the isolines and the top equations.

    *"Done"*

  - o Figure 11: Same as Figure 10, not 11.

*"Solved"*

- Line 80: AND in capital letters

  *"Done"*

- Section 5 title: 'future' to 'Future'

  *"Done"*

**REVIEWER 3**

The authors assessed the future changes of Dry-Hot compound events in Pyrenees. Based on the definition of the duration (D) and magnitude (M) of the dry-hot event, they analyzed the climatology of the two properties. D and M from climate model simulations were corrected and then used for future projection in the study area. Overall, this study falls within the scope of this journal. It could be improved by clarifying D/M definitions and bias correction methods/results. Several comments are as follows.

*"We gratefully acknowledge the reviewer's comments and the revision of our manuscript. The article has been revised in accordance with the referee's comments and suggestions, which are addressed below. Our responses appear in italic and quotation marks"*

Major comments:

(1)The definition of D, M, and EM is not quite clear. For example, the definition of the 90$^{th}$ percentile for both the D and M needs to be clarified.

*"We have rephrased L159-160 to make clearer the event definition. In addition, we have changed the 90$^{th}$ percentile by 95$^{th}$ percentile to gain sensibility when detecting extreme dry spells and heat extremes."*

(2) For the bias correction, the D and M are corrected directly. Some comments or comparisons with the multivariate bias correction of climate variables (and then derive the D and M) could add merit to this study.

*"The authors have made structural changes in the performance of the bias correction to guarantee more correct and transparent results in terms of the uncertainty of this type of correction in climate simulations. Below we summarize the changes made in the methodological aspect (integral restructuring of section 2.4) and to the assessment of the results of the application of the bias correction methods (section 4):"*

- o *"We first applied empirical quantile mapping (EQM) to the original temperature and precipitation data, rather than directly to D, M and EM. If the correction is performed for D, M and EM, the physical connection between the original precipitation and temperature can be omitted (i.e. the physical mechanisms are lost)."*

- o *"Subsequently, we incorporated a multivariate BC method, the Multivariate Bias Correction with N-dimensional probability density function transform (MBCn), proposed by Cannon (2018). This method enabled us to maintain the structural dependence between temperature and precipitation (see Fig. 7, section 4), which is relevant when working with compound events. Throughout section 4, the performance of MBCn is compared to EQM."*

- o *"To analyse the uncertainty in the estimation of dry spells in the climate models and in the subsequent correction, the D-statistic of the Kolmogorov-Smirnov test (L217-222) was used. Section 4 shows that the results are very irregular, which for the CANT region are acceptable, since the distribution simulated by the BC methods is close to the observed one; but this is not the case for the NMED region, where the distribution of the dry spells simulated and corrected by the two BC methods is clearly different according to the KS test. See Fig. 8."*

- *"Finally, the performance of the two BC methods in modelling daily temperature has also been analysed, with emphasis on the daily extreme values above p95 of Tx for each year (spring and summer), as well as on the daily extreme values of TX for each year within a dry spell. Section 4 explains that the EQM (denoted as UBC in the text (Univariate Bias Correction)) performs quite well, but when reaching the most extreme temperature values, those occurring within a dry spell, the MBCn is more accurate (Fig. 9 and Fig. S4)."*

- *"All these results suggest the need to employ MBCn when correcting future projections. This has led to some changes in section 5."*

(3) Presentations and discussions of patterns in several figures need to be more clear (e.g., Figure 6 and 12)

*"Figure 6 was removed because section 4 was fully restructured. Please review the new version of section 4. In addition, section 5 has also been improved and now the explanations are more concise. Please review the improved version of section 5."*

Other comments

Please correct typos in many places of the manuscript (e.g., line 34).

*"Done"*

Lines 139-140 (and Table 1):  how do you define the "90th percentile" for both D and EM? Is this threshold based on the temperature of MAM and JJA? Please clarify.

*"We added the following sentence: To ensure that independent and extreme spells were obtained, for each year (spring and summer, separately) we computed 95th percentile of dry spells duration and then we selected those with a duration greater than this threshold."*

In Table 1: There are multiple days with temperatures higher than $90^{th}$ percentile. How do you define the EM (average or maximum)?  Please make this clear.

*"Table 1 is about models. We defined EM in L159-163."*

Lines 157-162: This bias correction procedure is performed on the D and M. One can also adjust the climate variables and then compute the D and M. In this case, the multivariate bias correction is of particular interest to correct the dependence between the contributing variables of compound events (Cannon, A. J., 2018, Clim. Dynam; Zscheischler J. 2019 Earth System Dynamics Discussions). Some discussion or comparison on this would enhance this study.

*"Thanks for the comment. We have already addressed this issue in this review."*

Figure 6: For the corrected M and EM, the bias seems to be 0 for all seasons and regions. This means that the bias correction procedure has corrected almost all the systematic biases. Is this the case in Figure 6? Please explain or clarify the almost perfect performance of the correction.

*"The previous analysis did not get a clear view about uncertainties of the bias correction methods applies. We have fully restructured section 4.*

Figure 12: "The drivers of the three future periods of the compound event" The analysis of the driver is interesting. However, the estimation of the driver seems to be quite vague. How do you determine the driver? Please explain it clearly.

*"The drivers are drawn through the performance of CANT and NMED regions. The small sketches are intended to give an understanding of the patterns identified in the different scenarios and seasons. Most part of text has been rewritten to make these results more comprehensive."*

---

## Author Response (AR2)

**REVIEWER1**

General comments

In the revised version of this manuscript, the authors have addressed the major issue raised in the first review with respect to the bias correction of the duration of dry spells. Instead of applying quantile mapping directly to the duration pdf, they have applied a bias correction to the underlying time series before calculating duration. Additionally, they have also included an additional assessment of two multivariate bias correction methods. In their analysis, they assess changes in the duration of dry spells and temperatures during dry spells over the Pyrenees in future projections from an ensemble of regional climate models. Specifically, they assess changes in the annual mean duration of dry spells that exceed the annual 95th percentile of duration as well as changes in the annual mean extreme magnitude of dry spells, which is the annual mean of temperatures that exceed the 95th percentile of temperature during dry spells. They find that extreme temperatures generally increase during dry spells in future projections, while the changes in duration can vary depending on the assessed region. The analysis is clean and the results are well presented. There is also novelty in the presentation of changes in the bivariate distribution, which is a nice a feature of this paper. However, some clarifications are required in the text, particularly with respect to the event definition which I found created confusion when interpreting results. Furthermore, it would also help if some discussion was added on the model biases and the use of bias correction. With these minor changes, I would recommend this manuscript for publication.

*"We gratefully acknowledge the reviewer's comments and the revision of our manuscript again. The article has been revised in accordance with the referee's comments and suggestions, which are addressed below. Our responses appear in italic and quotation marks."*

Comments (P: page, L: Line)

P7 L161: I think the explanation of M and EM could be simplified as I found the description of EM and M as 'events' confusing. From what I understand, the event is the dry spell and this is has duration D which is a characteristic of the event. Then, M is the conditional distribution of temperatures during dry spells while EM is the conditional distribution of temperature during dry spells that exceed the 95th percentile of temperature. I believe changing the description of the calculation of the variables would make things easier for the reader. I would recommend removing any reference to M or EM as events.

*The authors agree with the reviewer's comment. In fact, it is an aspect that two of the reviewers emphasize that should be improved. For this reason, we have rephrased L161-166 and changed figure 2 to a real case in two different years.*

P7 L175-176: Why do you take the time series from one grid cell? Would it not be more consistent with observations if the mean of all grid cells in the region was taken?

*The EURO-CORDEX data have a resolution of ~11 km, while the starting grid has a resolution of 1 km. The regional averaging of the observational data is intended to smooth the observational data slightly, and thus avoid problems already known when using bias correction methods for downscaling (inflation problems in the corrected series -Maraun, 2013- and inability to generate daily subgrid variability).*

P8 L185-192: How do you treat zero values in this quantile mapping? Are the lowest precipitation wet days in the model simply converted to dry days?

*To correct the drizzle effect in the QM, we use the threshold of 1mm/day. We have added this*

*information in L200-201.*

P9 L211: Could you clarify what is meant by 'annually aggregated data'?
*We have rephrased it. Please see L220*

P9 L213: I think there is a comma missing after 'which was bias corrected'.

*Right, comma added.*

P16 L195-315: As is highlighted here by the authors, the results shown in Figure 8 show that the corrected CDFs are in many cases very similar to the uncorrected version or worse. Is there a value in using the 'corrected' dry spells vs. 'uncorrected'?

*The results show trivial improvements, as they transfer the intrinsic error of the model to the bias correction performance. Precisely, we have mentioned between lines 316 and 318 that the results should be taken with caution. We have discussed these results in the discussion section.*

P18 Figure 9: I think it would be informative to include a QQ plot of the uncorrected temperature also, to give an idea of the biases present in the temperature series.

*The authors are grateful for the reviewer's suggestion. We have included the QQ plot of the uncorrected temperature in the supplement (Fig. S4). Figure 9 of the manuscript allows to see the performance of the UBC and MBCn at the most extreme values.*

P19 L327: Could you clarify what is meant by 'magnitude of intervention' in the section title?

*This section title was quite confusing. We changed by: Future changes in the variables underlying the compound event*

P19 L329-335: Are these results shown for 'uncorrected' or 'corrected' time series? Which bias correction method is selected if the corrected time series are assessed?

*The reviewer is right. The authors forgot to mention that the BC method used to show the projected results was the MBCn. Please see L338-339*

P21 Figure 11: I assume this figure shows the multi-model mean projected change? If so, it would helpful to state this in the caption.

*We have stated the reviewer suggestion in this caption.*

P22 Figure 12: I think this figure still needs some clarification. Does each point in the scatter plot represent the multi-model annual mean of D and EM in a given year? The use of the notation D and EM in the caption is also quite confusing here. On P6 in the event definition, this notation is used to represent individual events or days (for M and EM), but here the notation is used to refer to annual mean anomalies of D and EM. This is also the case in Figure 10 and 11. I would suggest using different notation for individual events and annual mean values of those events.

*Yes, each point in the scatter plot represents the multi-model annual mean of D and EM in a given year. We have rephrased this figure caption, as well as the respective figure caption in the supplementary material, following your suggestions.*

Discussion section: The use of bias correction to correct the distribution of dry spells will simply take the least wet days and convert them to dry days. As noted in Maraun et al. (2017), this may correct biases resulting from the drizzle effect but not biases resulting from topographical issues or underestimation in the persistence of anti-cyclonic conditions. I think the authors should add some

discussion on this point as well as add some discussion on the results of their bias correction analysis and the performance of climate models in their representation of dry spell both before and after bias correction.

*The authors have added a discussion paragraph on the performance of bias correction in resolving bias in dry spells (temporal dependence). Please see lines 422-427.*

**Reviewer 2**

General comments

Thank you for the opportunity to review the revised version of the manuscript "Assessing internal changes in the future structure of Dry-Hot compound events. The case of the Pyrenees". The authors had put in considerable effort in addressing most of the aspects raised by the three reviewers, particularly in terms of the regionalization procedure, the event definition, and the multivariate bias correction. During the review process the authors have also found that they were estimating D, E and EM events in Figs. 3 and 4 for the full period 1981-2015 and not annually, and hence, only Figure 1 has not been changed from the originally submitted manuscript. In addition, the authors have added to the revised manuscript a new figure (revised version Figure 5), that was not requested by any of the reviewers, but it is my view that providing an illustration of the large-scale drivers of the compound event in analysis is an added value to the manuscript and that it supports the results. In general, I consider that most of the reviewers' comments were addressed and that the key necessary changes were performed to the manuscript, greatly improving its original version. However, I still have minor comments to make, which I outline with more detail below, and I leave the decision on these final suggestions to the journal editor, if I may.

*"We gratefully acknowledge the reviewer's comments and the revision of our manuscript again. The article has been revised in accordance with the referee's comments and suggestions, which are addressed below. Our responses appear in italic and quotation marks"*

**Specific comments**

Figure 2 - In the reply to my suggestion to improve Figure 2 with an illustration of an observational year, the authors considered that the scheme is enough to illustrate the procedure. However, I still think that it was a common point to all the reviewer's comments that the definition of the event required clarification. In addition, as an answer to a Reviewer's comment the authors have exemplified with the particular year 1981 of the grid cell i,j, where there were 2 dry spells accounting for 15 and 20 days, respectively. In this way, we agree that the exemplification with an observational year helps to better understand the procedure, and this would be a good way to improve Figure 2 and the clarification of the event definition.

*The authors have considered the reviewer's suggestion. In this sense, an example of the detection of the extreme dry spells (D), the conditional distribution of temperatures during dry spells (M) and the conditional distribution of temperature during dry spells that exceed the 95th percentile of temperature (EM), in a cell i,j has been included in the manuscript as an example. In this sense, figure 2 shows the time series of daily maximum temperature and daily precipitation for the years 1995 and 2006 for the range x = 2.9, y = 42.5, thus illustrating the procedure followed in this work.*

The authors could also mention that they performed a sensitivity analysis using the threshold 90th, and briefly outline the main differences in the analysis using a slightly lower threshold.

*We have mentioned that we performed a sensitivity test to select the adequate threshold. The selection of the 95th percentile allows us to obtain sufficiently long and robust dry spells in the most humid areas of the Pyrenees. Please see L.161-162*

P2 L43 - I'm afraid the authors have misunderstood my suggestion to include a short introduction in one sentence to the definition proposed by Manning et al (2019), as this work is an extension of the definition proposed by Manning et al (2019). I was not suggesting to just removing the parenthesis to the reference, but to guide better the reader of the introduction that

the magnitude of a Dry-Hot event concerns the temperature and that the duration of a Dry-Hot concerns the length of the dry spell, as proposed by Manning et al (2019). This is not an obvious definition, in my point of view, and when reading the introduction for the first time the concept could be better addressed.

*We have rechecked your comment, but we think that it is already clearly stated that the magnitude of a Dry-Hot events refers to the temperature while the duration of these refers to the dry spells:*

*"although in Europe the magnitude (temperature) of these events was revealed to have greater weight than their duration (dry spells) as indicated by Manning et al., (2019)."*

Moving averages and future periods - I suggested the authors to explain the use of a 7-year moving average and the authors changed for 5-year moving year average without explaining. The authors could mention that in addition to the 5-year moving year average, they also used a 7-year moving average, and which were the main differences, if any.

*There is no difference between 5-yr and 7-yr for the running window. We have used an odd number because there must be a central year; also 5-yr is half a decade.*

Moreover, the authors have changed the analyzed future period from 2011-2100 to 2006-2100 without mentioned it in the review, and the three periods changed from (2011-2040, 2041-2070 and 2071-2100) to (2016-2035, 2046-2065 and 2081-2100). This should be stated clearer. The new future periods have now 19 years each (instead of 29 as before), and I think that the correct period to include in the abstract L13 is 2016-2100 instead of 2006-2100.

*The authors would like to apologize for this oversight. The changes were marked in the text but not mentioned in the revision. The selection of these new periods is in accordance with the periods used in fifth IPCC (Stocker, 2015). The data from the RCPs starts in 2006. We used the complete series in Fig 10 and 11.*